bioengineering

gait stability, dynamic walking, robustness, Lyapunov exponent, stability measure, fall prediction

**Author for correspondence:**
Sjoerd M. Bruijn
e-mail: s.m.bruijn@vu.nl

# The validation of new phase-dependent gait stability measures: a modelling approach

Jian Jin[1], Dinant Kistemaker[1], Jaap H. van Dieën[1], Andreas Daffertshofer[1,2] and Sjoerd M. Bruijn[1,2,3]

[1]Department of Human Movement Sciences, Faculty of Behavioural and Movement Sciences, Vrije Universiteit Amsterdam and Amsterdam Movement Sciences, Amsterdam, The Netherlands
[2]Institute of Brain and Behavior Amsterdam, Amsterdam, The Netherlands
[3]Orthopaedic Biomechanics Laboratory, Fujian Medical University, Quanzhou, Fujian, People's Republic of China

JJ, 0000-0003-2525-0471; JHvD, 0000-0002-7719-5585;
AD, 0000-0001-9107-3552; SMB, 0000-0003-0290-2131

Identification of individuals at risk of falling is important when designing fall prevention methods. Current measures that estimate gait stability and robustness appear limited in predicting falls in older adults. Inspired by recent findings on changes in phase-dependent local stability within a gait cycle, we devised several phase-dependent stability measures and tested for their usefulness to predict gait robustness in compass walker models. These measures are closely related to the often-employed maximum finite-time Lyapunov exponent and maximum Floquet multiplier that both assess a system's response to infinitesimal perturbations. As such, they entail linearizing the system, but this is realized in a rotating hypersurface orthogonal to the period-one solution followed by estimating the trajectory-normal divergence rate of the swing phases and the foot strikes. We correlated the measures with gait robustness, i.e. the largest perturbation a walker can handle, in two compass walker models with either point or circular feet to estimate their prediction accuracy. To also test for the dependence of the measures under state space transform, we represented the point feet walker in both Euler–Lagrange and Hamiltonian canonical form. Our simulations revealed that for most of the measures their correlation with gait robustness differs between models and between different state space forms. In particular, the latter may jeopardize many stability measures' predictive capacity for gait robustness. The only exception that consistently displayed strong

correlations is the divergence of foot strike. Our results admit challenges of using phase-dependent stability measures as objective means to estimate the risk of falling.

## 1. Introduction

Falling is a major threat, especially for older adults. About one-third of all adults older than 65 years fall at least once per year, often with serious injuries and fractures as consequences [1]. Without a doubt, there is an urgent need to identify individuals at risk, in particular, when it comes to customizing fall prevention [2]. Various measures have been proposed to quantify the risk of falling in humans. Given the apparent relation to dynamic stability, the local divergence exponent (maximum finite-time Lyapunov exponent) [3,4] and the maximum Floquet multiplier [5] are the first to mention. Lockhart & Liu [6] and Toebes *et al.* [7] suggested that the local divergence exponent may allow for discriminating fallers from non-fallers, with accuracy of 80% at best [8]. Although 80% accuracy is a fair achievement in view of the complexity of the dynamics involved in human walking, it may still lead to a large number of false positives (predicted fallers that are not prone to fall) and—arguably worse—false negatives (predicted non-fallers that are prone to fall) of the elderly population. Therefore, a higher accuracy in fall prediction is of great importance.

Most stability measures are obtained by looking at either stride-to-stride deviations of an isolated point (e.g. heel strike state) or at average deviations over all phases of gait cycles. Humans, however, have phase-dependent gait stability, as we display distinct responses to perturbations when applied at different phases of a gait cycle [9,10]. Estimating gait stability at an isolated point of or from an average over the gait cycle may thus fail. Norris *et al.* [11] identified the phase-dependency in local stability using a 'simple' walking model, and Ihlen *et al.* [12] reported intra-stride local stability variations in human gait. These observations and studies ascertained the potential for using phase-dependent local stability information to quantify gait stability.

While the term *gait stability* may be intuitively sound, it is often confused with *gait robustness*. Robustness can be operationalized as the maximum magnitude of perturbations that a walker can handle, while stability reveals whether or not a walker will return to its periodic motion after an infinitesimal perturbation. Infinitesimally implies locality (here close to a limit cycle). Stability, or better, *local stability*, does, in general, not relate to the maximum *size* of the perturbation that a nonlinear system can handle, i.e. robustness. As such, the relation between a given stability measure and gait robustness is far from straightforward. Take the maximum Floquet multiplier as an example: it is a perfectly valid stability measure for periodic systems [13] but modelling [14–16] as well as experimental studies [17] showed its limitations as proxy for gait robustness.

In a limit cycle system like a passive dynamic walker, locally stable phases and unstable phases may coexist. In that case, averaging local stability measures may lead to a loss of information regarding local stability tendencies. Perturbations applied during unstable phases may lead to larger deviations from the limit cycle than applied during stable phases. To address this, phase-dependent stability measures have been introduced by Ali & Menzinger [18]. By and large, they are based on a limit cycle's local stability quantified by the aforementioned responses to infinitesimal perturbations, but applied at different phases of the cycle. When it comes to human gait, phase-dependent stability measures have, in fact, been shown to potentially improve the accuracy of fall prediction over cycle-averaged parameters [19]. Still, only a limited number of studies [12,19–22] on human gait used such phase-dependent stability measures, and further validation is needed.

The goal of the current study was to further validate phase-dependent stability measures for walking. Directly testing these stability measures on empirical data is hard, given the enormous complexity and variability of human walking. Therefore, rather than directly testing on empirical data, we asked: Can phase-dependent stability measures provide proper predictions of gait robustness in 'simple' walking models [23]? A positive answer will encourage application to experimental data, while a negative answer should be considered a call for alternative approaches. To answer our question, we devised several phase-dependent stability measures based on the stability analysis of Garcia [24], Goswami *et al.* [25] and Norris *et al.* [11] applied to a two-dimensional compass walker model with point feet [11,24,25]. To validate our stability measures, we determined their relations with the walker's gait robustness. We also tested whether these relations were stronger than those between the commonly used local divergence exponent and gait robustness. Since we aimed to test the theoretical merits of phase-dependent stability measures, we here circumvent the inclusion of time-series analysis method.

Hence, we did not apply the method proposed by Ihlen *et al.* [12,21]. Next, we asked whether the corresponding results transferred to a slightly more complicated model including circular rather than point feet. Finally, we investigated the extent to which our phase-dependent stability measures are invariant to different state space formulations, in particular, when formulating the walker dynamics in either Hamiltonian canonical or Euler–Lagrange form. A dependency of the choice of state space form would severely limit the usefulness of any measure for fall prediction in humans.

# 2. Methods

## 2.1. Models

### 2.1.1. Compass walker with point feet

The first model we used is illustrated in figure 1. It is a passive dynamic compass walker with point feet [11,24,25]. In brief, the model consists of two massless legs connecting the hip point mass $M$ and foot point masses $m$; we denote the mass ratio as $\beta = m/M$. The state variables are defined as $s = (\theta, \varphi, \dot{\theta}, \dot{\varphi})^T$, where $\theta$ represents the angle of the stance leg with respect to the normal of the inclined plane, $\varphi$ is the angle between the legs, and $\dot{\theta}$ as well as $\dot{\varphi}$ are the respective angular velocities. A gait cycle (one step) comprised a swing phase and an instantaneous double stance phase. At the double stance moment (= foot strike), we assumed a fully inelastic collision of the leading leg and exploited the conservation of angular momentum that gives rise to a linear mapping from pre- to post-collision state. The model walked down a surface with slope $\gamma$. Throughout our paper, we rescaled time by $\sqrt{l/g}$ to ease legibility. A more detailed description including the Euler–Lagrange equations of motion can be found in appendix A, where we adopted the notation of Garcia [24] and Norris *et al.* [11].

We also analysed this compass walker model using Hamiltonian form. In that formulation, the state variables were defined as $s = (\theta, \varphi, p_\theta, p_\varphi)^T$, where $p_\theta$ and $p_\varphi$ denote the canonical conjugate momenta with respect to $\theta$ and $\varphi$. The corresponding equations of motion can be found in appendix B. In order to find the general relations between stability measures and gait robustness of the model, we included a variety of configuration parameters by systematically varying the slope $\gamma$ and mass ratio $\beta$ ($\gamma = 10^{-3} \ldots 1.2 \cdot 10^{-2}$ in steps of $\Delta\gamma = 2 \cdot 10^{-4}$ and $\beta = 2 \cdot 10^{-3} \ldots 1 \cdot 10^{-1}$ in steps of $\Delta\beta = 2 \cdot 10^{-3}$), yielding $56 \times 50 = 2800$ parameter combinations.

### 2.1.2. Compass walker with circular feet

We also simulated a two-dimensional compass walker model with circular feet. In this model, the state variables are $s = (\varphi_1, \varphi_2, \dot{\varphi}_1 \dot{\varphi}_2)^T$, where the definitions of these angles are given in figure 2; more details of the configurations and equations of motions can be found in Wisse & Schwab [26]. To obtain the equations of motion and collision equations, we followed the approach by Casius *et al.* [27]. The circular feet walker also has instantaneous foot strike. The foot radius $r$ was varied as $r = 0.01 \ldots 0.49$ in steps of $\Delta r = 0.04$ and the slope $\gamma$ were varied like for the point feet walker model yielding $13 \times 56 = 728$ parameter combinations.

## 2.2. Numerical simulations

For the point feet walker, we used a Runge–Kutta (4,5) integrator, while for the circular feet walker, we used an integrator for non-stiff differential equations; all simulations were realized in Matlab (The Mathworks, Natick, MA, USA). The absolute tolerance and relative tolerance were both $10^{-8}$ for the point feet walker and $10^{-10}$ for the circular feet walker. To avoid foot scuffing, foot strike detection condition was defined by constraining $\theta$ to be less than $-0.05$ radians for the point feet walker, and the swing leg to be in front of the stance leg and the inter-leg angle to be larger than $5°$ for the circular feet walker.

For every parameter combination, we searched for a period-one solution (i.e. a solution where the states at the end of the step are the same as those at the beginning of the step) using the Newton–Raphson method [26]. Whenever a period-one solution was detected, we determined the Floquet multipliers from the Poincaré map reflecting error multiplication factors from the step-to-step map following [26]. We ignored all the unstable solutions (and thus all the short-period gaits; [28]) in further calculations by removing all the solutions with a maximum Floquet multiplier with a modulus greater than 1.

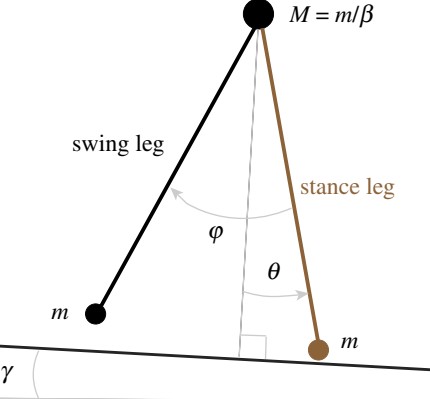

**Figure 1.** Compass walker model with point feet. This model consists of two massless legs connecting the hip point mass $M$ and foot point masses $m$. $\theta$ is the angle of the stance leg with respect to the normal of the inclined plane and $\varphi$ is the angle between the legs.

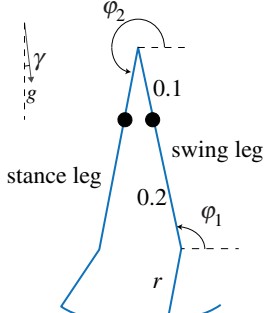

**Figure 2.** Circular feet compass walker and its configurations. A snapshot is captured at the end of a step. For simulation purposes, gravity was tilted to simulate an inclined slope $\gamma$. $\varphi_1$ and $\varphi_2$ are the angles of the swing leg and stance leg with respect to the inclined slope; $r$ is the radius of the foot.

## 2.3. Stability measures

### 2.3.1. Gait robustness

In order to assess the robustness of the models, we determined the maximal step-up and step-down perturbation the two models could handle for every parameter combination. To this aim, the walker started from its stable period-one solution after which, in the first step, we applied a one-time step-up/step-down floor height difference, see figure 3 for an illustration of these perturbations. To generalize gait robustness in the light of another type of perturbation, we also applied a one-time constant push or pull at the centre of mass of each leg. We performed this type of perturbation only for the circular feet walker. The push or pull was applied during the first step for 0.1 s, starting at the moment the hip angle was zero. Gait robustness was then quantified as the sum of the maximum allowable push and pull perturbation or the sum of the maximum step-up and step-down perturbation without falling. We gradually increased this perturbation size (with precision up to $10^{-5}$ for the point feet walker, $10^{-4}$ for the circular feet walker with step variation perturbation and $10^{-2}$ with the push and pull perturbation) until, after the perturbation, the model was no longer able to complete 30 steps (which were empirically enough for the disturbance to be attenuated if the walker would not fall earlier).

### 2.3.2. The local divergence exponent

To determine the widely used *local divergence exponent* or short-term maximum finite-time Lyapunov exponent, we employed Rosenstein's algorithm [29]. In brief, per parameter combination, we performed five simulations of 200 steps where at each step, a floor height variation was added (zero-centred Gaussian white noise with a standard deviation of $2 \cdot 10^{-5}$). The time series of full states of the model were the direct input of the Rosenstein's algorithm for constructing the corresponding state

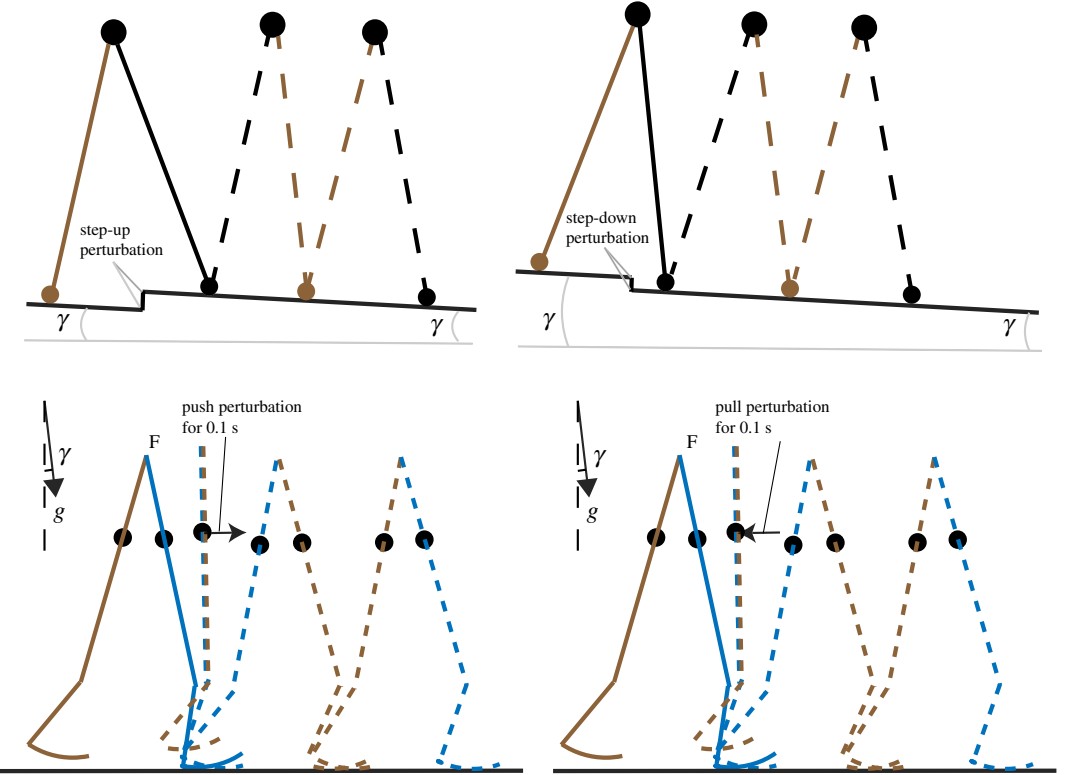

**Figure 3.** A schematic illustration of step-up/step-down perturbation and push/pull perturbation. The floor height perturbation is added only at the first step, and push/pull perturbation is added upon hip angle reaches zero and lasts for 0.1 s during the first step.

space. Then, we applied Rosenstein's algorithm [29,30] to estimate the local divergence exponent for each of these simulations, and averaged over five simulations per configuration.

### 2.3.3. Phase-dependent stability measures

#### 2.3.3.1. Swing phase

To quantify the local stability of the swing phase of our models, we assessed how their continuous dynamics, $\dot{s} = f(s)$, behave in the presence of an infinitesimal perturbation $\delta(t)$. This behaviour can be approximated by linearization using the (time-dependent[1]) Jacobian at every point along the period-one solution

$$\dot{\delta}(t) = J(t) \cdot \delta(t) . \tag{2.1}$$

Note that $J(t)$ is not explicitly dependent on time (both walkers are time-invariant systems), but implicitly through $\delta(t)$. The eigenvalues of $J(t)$ quantify the rate at which the system returns to (negative eigenvalue) or moves away from (positive eigenvalue) the period-one solution in the corresponding eigen-directions after infinitesimal perturbations. Here, we would like to add that these eigen-directions may be (partly) along the period-one solution. In that case, the eigenvalues belonging to these eigenvectors quantify how perturbations yield a phase shift. Since such phase shifts in our model do not result in moving away from the period-one solution, they do not affect the stability of the period-one solution as a whole, and hence do not provide useful information about gait stability or gait robustness. One can circumvent this by performing a coordinate transformation to obtain eigenvalues that belong to eigenvectors that are orthogonal to the period-one solution.

Following Ali & Menzinger [18] and Norris *et al*. [11], we used a moving coordinate frame $U(t)$, with one axis always remaining tangent to the period-one solution $f(s(t))^T \cdot \| f(s(t)) \|^{-1}$, while the other axes span a hyperplane orthogonal to the period-one solution; cf. Figure 4. One can use this moving coordinate frame to transform perturbations from the global frame $\delta(t)$ to the moving coordinate

---

[1]Although local stability is defined in terms of time, for a period-one solution, this directly translates to phase.

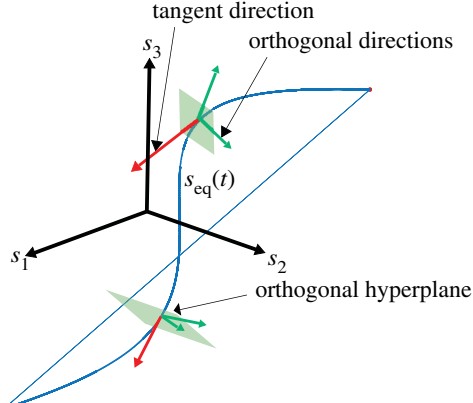

**Figure 4.** State space and moving coordinate frame. The period-one solution is displayed as the blue curve with a jump occurring at foot strike. We expressed our stability measures in the orthogonal hyperplane since we are only interested in the eigenvalues that belong to the eigenvectors that are orthogonal to the trajectory (i.e. the green arrows lying in the green plane), and can ignore eigenvalues that belong to eigenvectors that are along the period-one solution (i.e. red arrow perpendicular to the green plane). The blue straight line is due to the switch between swing leg and stance leg at foot strike.

frame $\delta'(t)$ following

$$\delta'(t) = U(t) \cdot \delta(t) . \tag{2.2}$$

By taking the derivative of equation (2.2) with respect to time and using equation (2.1), one can obtain

$$\dot{\delta}'(t) = \left( U(t) \cdot J(t) \cdot U(t)^{-1} + \frac{\mathrm{d}U(t)}{\mathrm{d}t} \cdot U(t)^{-1} \right) \cdot \delta'(t) =: \bar{J}(t) \cdot \delta'(t) , \tag{2.3}$$

in which $\bar{J}(t)$ describes how perturbations evolve. However, due to our coordinate transform, the perturbations $\delta'(t)$ in the tangent and orthogonal directions are uncoupled. That is, the $\bar{J}(t)$ matrix obeys the form

$$\bar{J}(t) = \begin{bmatrix} \lambda_{\|\|}(t) & \lambda_{\|\perp_1}(t) & \cdots \\ \lambda_{\perp_1\|}(t) & \lambda_{\perp_1\perp_1}(t) & \cdots \\ \vdots & \vdots & \ddots \end{bmatrix} , \tag{2.4}$$

where vanishing (zero-valued) $\lambda_{\perp_i\|}(t)$ imply that a tangent initial perturbation does not evolve onto the orthogonal hyperplane [11]. Given that phase shifts do not alter stability in our model, one creates a reduced Jacobian matrix $J'(t)$ by removing the first column and top row of matrix $\bar{J}(t)$. The eigenvalues of $J'(t)$ describe the rate at which infinitesimal perturbations return to (negative eigenvalue) or move away (positive eigenvalue) from the period-one solution in the corresponding eigen-directions, which are all orthogonal to the direction of the period-one solution, i.e. they do not quantify a phase shift.

Figure 5 illustrates the evolution of all eigenvalues of the reduced Jacobian $J'(t)$ during the swing phase for the point feet walker with an arbitrary $\gamma$ and $\beta$. These eigenvalues indicate phase-dependent local stability orthogonal to the period-one solution, and one may derive potentially useful measures from them. To do so, one can calculate the *trajectory-normal divergence rate, $\Gamma(t)$*, as the sum of all eigenvalues of $J'(t)$, indicating the mean contraction or expansion rate of all neighbouring perturbations rather than along a single eigendirection (figure 5, blue curve).

The advantages of using the trajectory-normal divergence rate instead of the maximum eigenvalue of the local Jacobian matrix to quantify local stability are threefold: (i) it quantifies the growth of all perturbations around the unperturbed trajectory rather than in a single eigendirection; (ii) it removes dimension that corresponds to the tangential; (iii) it is invariant against rotation of the normal coordinates since the growth of perturbations is evaluated only in the orthogonal space that is always normal to the time-varying tangential. From this trajectory-normal divergence rate, we calculated the following three measures.

*Stability measure i.* The *maximum of trajectory-normal divergence rate* during the swing phase, i.e. $\max_{t \in \mathrm{Swing}} \Gamma(t)$, quantifies the point with the largest trajectory-normal divergent rate in the swing phase. The higher the value, the more deviation an infinitesimal perturbation at this phase will yield from

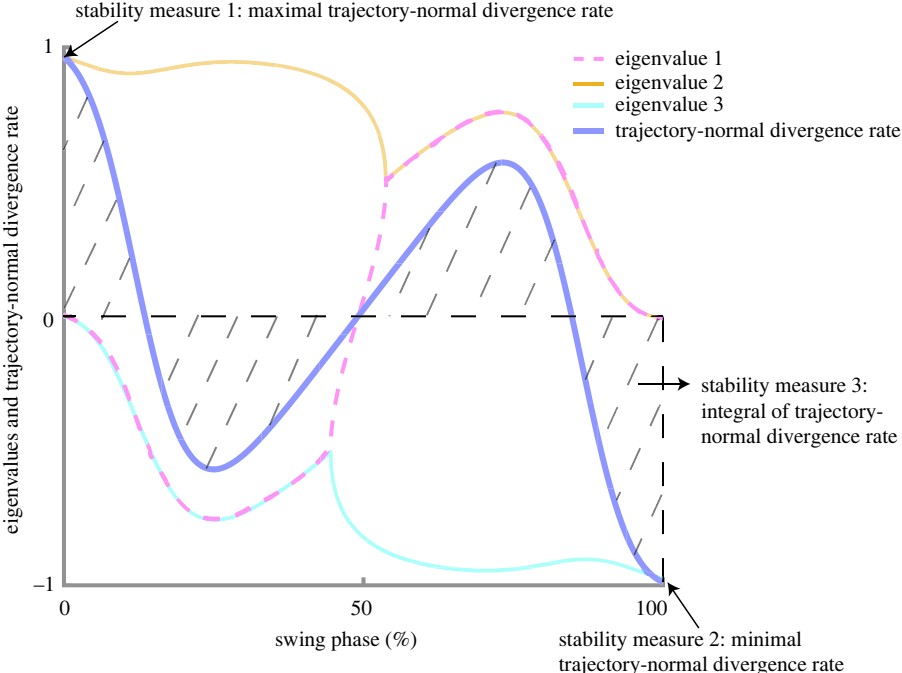

**Figure 5.** Eigenvalues and trajectory-normal divergence rate over the swing phase for the point feet walker. We indicate how the three stability measures are derived from the trajectory-normal divergence rate (i.e. blue bold curve, obtained by taking the sum of all three eigenvalues). A similar figure could be made for the circular feet walker, but the shape of the curves will differ.

the orbit. Since it is a nonlinear system, it remains to be shown if for the walker models analysed here, a higher value of this measure would result in a lower gait robustness.

   *Stability measure ii.* The *minimum of the trajectory-normal divergence rate* during the swing phase or $\min\limits_{t \in \text{Swing}} \Gamma(t)$ quantifies the point with smallest trajectory-normal divergent rate in the swing phase. The higher this value, the more deviation an infinitesimal perturbation at this phase will yield from the orbit; cf. measure *i* above.

   *Stability measure iii.* The *integral over the trajectory-normal divergence rate* over the swing phase, $\int_{t \in \text{Swing}} \Gamma(t) \mathrm{d}t$, provides the long-time multiplication factor of the volume of all neighbouring infinitesimal perturbations over the swing phase. Similar to the previous measures, the higher this value, the more the infinitesimal perturbations will cause on average a deviation from the orbit. Note that this measure was also studied in the physics literature where it is known as the *trajectory-normal repulsion rate* $\rho_T$; it indicates the normal growth of infinitesimal normal perturbations to the trajectory over the time interval $[0, T]$ [31,32].

### 2.3.4. Foot strike and gait cycle

A complete gait cycle (one step) of the compass walker contains a swing phase and an instantaneous double stance event (foot strike). Next to quantifying the local stability of the swing phase of the period-one solution of our models, we also quantified the local stability of the foot strike. We did this in a similar way as for the continuous dynamics. That is, we determined the Jacobian $J_{FS}$ of the foot strike event. Its eigenvalues quantify the degree of deviation a perturbation may cause during the discrete event (i.e. eigenvalue modulus greater than 1 indicates instability[2]). Similar to the continuous case, where we eliminated the phase shift dimension before estimating the trajectory-normal divergence rate, we finally derived the *divergence* as product of all non-vanishing eigenvalues. Since the foot strike event can be considered an important phase in the gait cycle, we also considered its divergence as a phase-dependent stability measure.

   *Stability measure iv.* The *divergence of the reduced Jacobian of the foot strike*, div $J'_{FS}$, quantifies the multiplication factor of the volume of all infinitesimal perturbations surrounding the period-one solution before and after the foot strike event. The closer this measure is to zero, the more likely it is that the perturbations after foot strike is attenuated.

---

[2]This is different from continuous systems, where the maximum eigenvalue $\lambda > 0$ indicates local instability

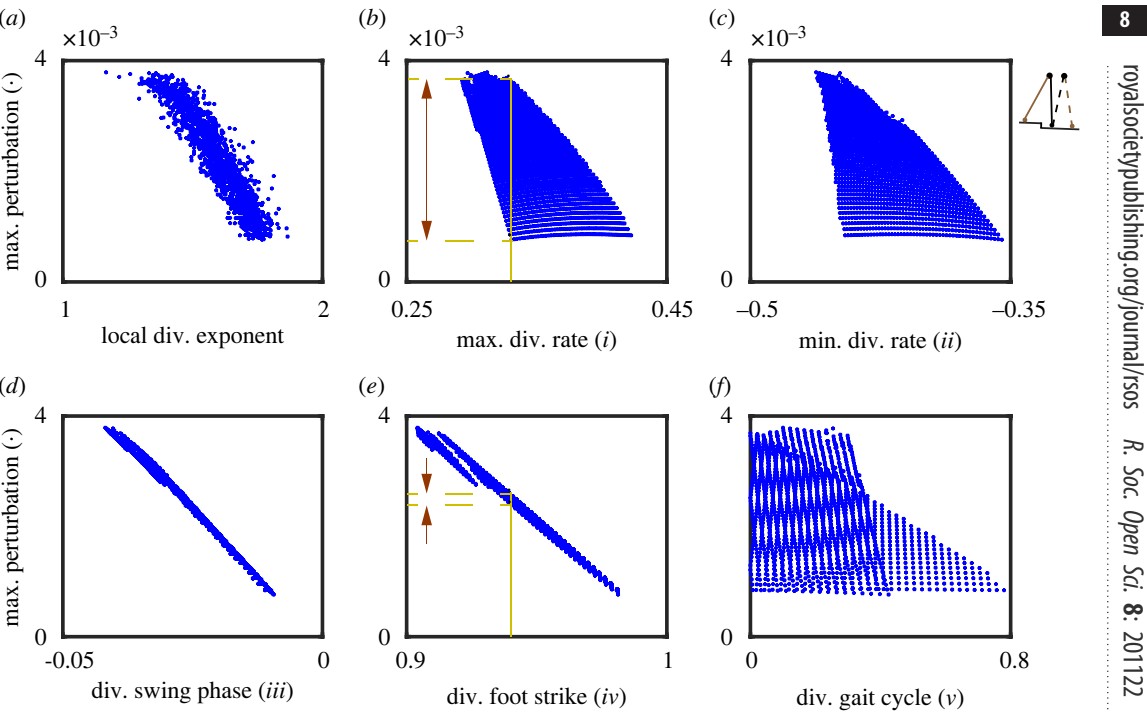

**Figure 6.** Relation between $1 + 5$ stability measures and gait robustness (maximum perturbation the model can handle) for the point feet walker represented in the Euler–Lagrange form. (*a*) The local divergence exponent, (*b–f*) phase-dependent stability measures *i* to *v*, respectively. Every data point represents a parameter combination of the model. (*b,e*) Give two examples of a 'bad' and a 'good' correlation. For a given value of stability measure, measure *i* predicts a wide range of maximum perturbation (bad), while measure *iv* predicts accurately only a small range of maximum perturbation (good).

### 2.3.5. Floquet multipliers

*Stability measure v.* The *divergence of a full period-one solution* can be obtained by taking the absolute value of the product of all non-vanishing Floquet multipliers. Although it is known that the maximum Floquet multiplier is not an optimal proxy for gait robustness [17], the divergence of a gait cycle could be an alternative for further validation.[3]

## 3. Results

In order to evaluate how well these phase-dependent stability measures and the local divergence exponent predict gait robustness, we correlated them with gait robustness (= maximum perturbation the model could handle), given our sets of parameter combinations. Before selecting our correlation measure, the general rubric for a 'good correlation' maintains that (i) for a given stability measure value, there should be a small variation in the corresponding gait robustness; (ii) a monotonic increasing stability measure should be able to predict a monotonic increasing (positive correlation) or decreasing (negative correlation) gait robustness. Two simple examples of a 'good' and a 'bad' correlation are illustrated in figure 6*b,e*. Instead of using Pearson correlation coefficient which cannot quantify nonlinear associations, we used Kendall rank correlation as summarized in table 1. This non-parametric correlation coefficient (between −1 and 1) evaluates the similarities in the ordering of the data when ranked by two variables. A coefficient value of 1 (−1) satisfies the monotonic relation that when one variable increases, the other variable increases (decreases); a coefficient value of 0 implies the absence of any association between the two variables. We considered the Kendall rank correlation coefficients larger than 0.7 (or smaller than −0.7) to be indicative for strong correlations. For truly promising stability measures predicting the robustness of a human gait, however, we expect the correlation coefficients to be larger than 0.9 (or smaller than −0.9).

[3]All data and codes can be found via under https://datadryad.org/stash/share/sXS8kl3SiaW1yteinh7-Tjuq5UP5iU5-pIu0lTbNGl8.

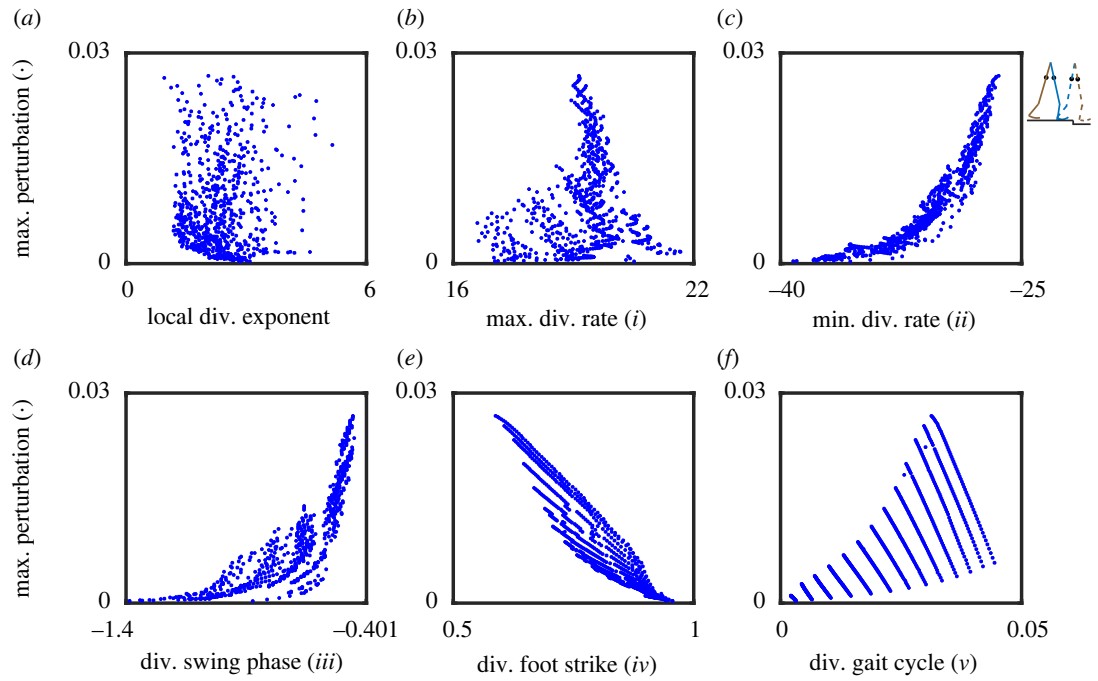

**Figure 7.** Relation between 1 + 5 stability measures and gait robustness (quantified by the sum of maximum allowable step-up and step-down perturbation) for the circular feet walker. (*a*) The local divergence exponent and (*b*–*f*) phase-dependent stability measures *i* to *v*, respectively. Every data point represents a parameter combination of the model.

**Table 1.** Kendall rank correlation coefficients of stability measures with gait robustness for the different models, different perturbation types and different state space forms. Green shading indicates negative correlations (higher stability measure value corresponding to lower robustness, as theoretically expected) and grey shading indicates positive correlations. We considered the Kendall rank correlation coefficients larger than 0.7 (or smaller than −0.7) to be strong correlations and displayed them in bold font in the table. All of these strong correlations are statistically significant ($p < 0.001$).

| model | perturbation | local div. exponent | max. div. rate (*i*) | min. div. rate (*ii*) | div. swing ph. (*iii*) | div. foot strike (*iv*) | div. gait cycle (*v*) |
|---|---|---|---|---|---|---|---|
| $1_{Euler}$ | step-up/down | **-0.79** | -0.47 | -0.46 | **-0.97** | **-0.96** | -0.25 |
| $1_{Hamiltonian}$ | step-up/down | -0.54 | **-0.93** | 0.61 | **-0.93** | **-0.96** | -0.25 |
| $2_{Euler}$ | step-up/down | -0.01 | 0.05 | **0.87** | **0.74** | **-0.80** | 0.57 |
| $2_{Euler}$ | push/pull | 0.13 | -0.003 | **0.84** | 0.67 | -0.69 | 0.62 |

We illustrate this further in a series of figures, where we show the findings for the point feet walker (figure 6), the circular feet walker under step-up/step-down perturbation (figure 7) and under push/pull perturbation (figure 8), and provide a comparison between stability estimates for the point feet walker in Euler–Lagrange and Hamiltonian form (figure 9).

For the point feet walker, two measures (measure *iii* in figure 6*d* and measure *iv* in figure 6*e*) showed good correlations with gait robustness (figure 6), better than those of the local divergence exponent, suggesting that these phase-dependent stability measures might be useful for predicting gait robustness. In particular, the divergence of swing phase (measure *iii*, the integral of the trajectory-normal divergence rate, figure 6*e*) and the divergence of foot strike (measure *iv*, figure 6*f*) showed very good correlations.

Unlike for the point feet walker, for the circular feet walker model, the local divergence exponent and most of the phase-dependent stability measures were not correlated well with gait robustness (measure *i*, *iii* and *v* in figures 7 and 8). One exception was the minimal trajectory-normal divergence rate (measure *ii*

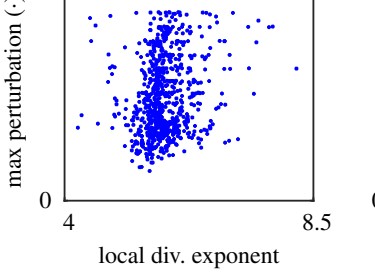
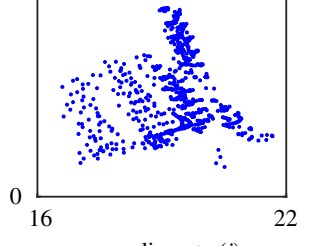
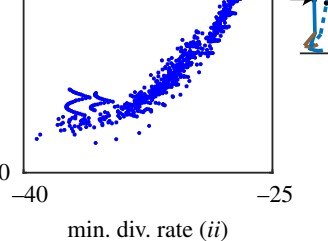
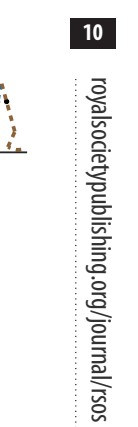
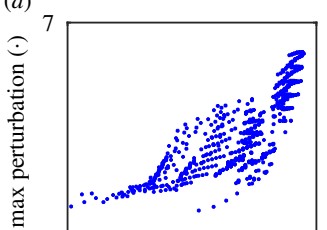
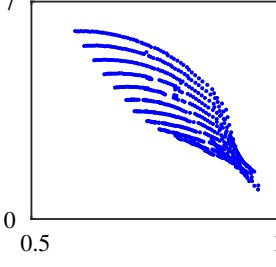
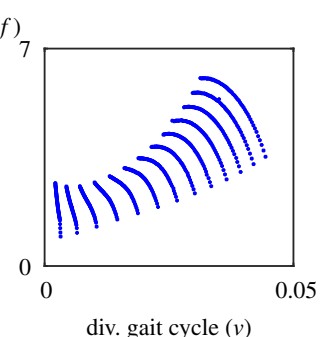

**Figure 8.** Relation between $1 + 5$ stability measures and gait robustness (quantified by the sum of maximum allowable push and pull perturbation) for the circular feet walker.

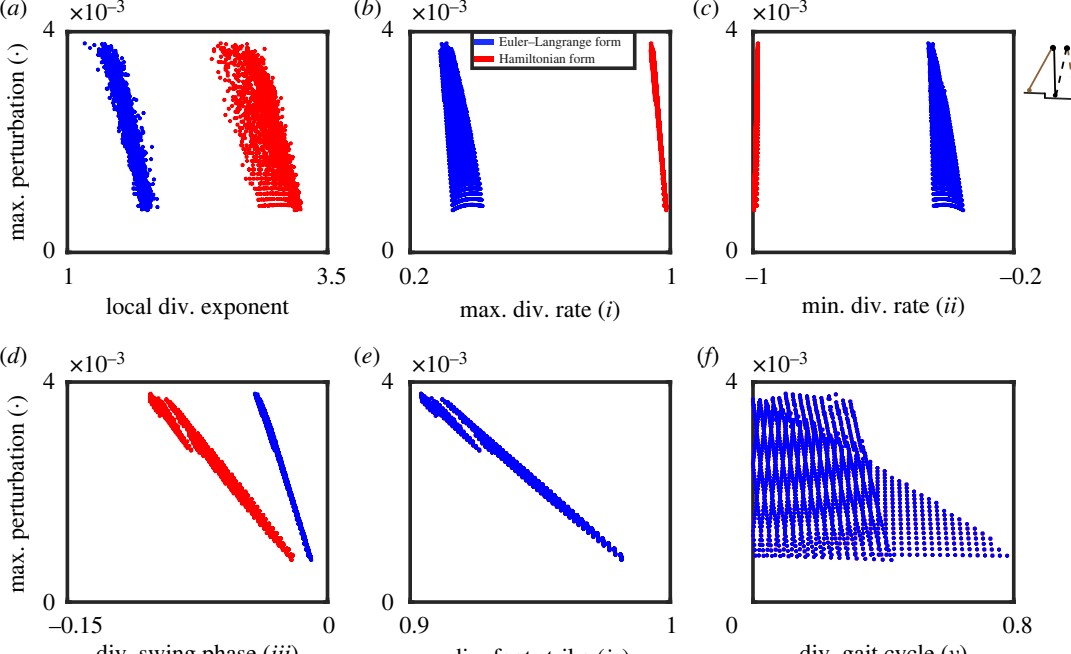

**Figure 9.** Comparing the relation between $1 + 5$ stability measures and gait robustness for the point feet walker in two different state space forms (red, Hamiltonian canonical form; blue, Euler–Lagrange form). The latter is identical to figure 7, but the axes are scaled differently to highlight the difference with the outcomes in Hamiltonian form.

in figures 7c and 8c) that, opposite to what we expected, was positively correlated with gait robustness. The inconsistent correlation directions of this measure *ii* cannot solely be explained by the measure itself but probably stem from the nonlinearity of different models. Only the divergence of foot strike (measure *iv* in figures 7e and 8e) showed an acceptable and consistent correlation with gait robustness, though

the correlation became slightly weaker from −0.797 (figure 7) to −0.688 (figure 8). Interestingly, applying different types of perturbations to quantify gait robustness (maximum step-up/step-down perturbation in figure 7 and maximum push/pull perturbation in figure 8) did not alter the general relations between stability measures and gait robustness, at least not qualitatively.

The influence of state space choice (here the Euler–Lagrange versus Hamiltonian form of the point feet walker) on the correlations between our stability measures and gait robustness is shown in figure 9. Note that the effect of (maximal) perturbations is independent of the state space choice. Furthermore, note that the figure 9e,f is necessarily the same for these two different coordinate systems. As can be readily seen from the figure, for the majority of measures, there was a clear dependency on the choice of state space form—see also appendix C for more mathematical details. This means that the choice of the state space form (e.g. whether we choose angular velocity or angular momentum as our state variable) will affect the values of phase-dependent stability measures and how well they correlate to gait robustness.

All in all, our results showed that (i) comparing figures 6 and 7, the correlations between phase-dependent stability measures and gait robustness highly depend on the walker model we considered; except for the divergence of foot strike (measure *iv,* figures 6e and 7e), all other measures show different correlations; (ii) comparing figures 7 and 8, robustness quantified by different types of perturbations (step-up/step-down versus push/pull) does not change the general relations between stability measures and gait robustness; (iii) comparing different state space forms in figure 9, the choice of state space form affects the values of phase-dependent stability measures and their correlations with gait robustness.

# 4. Discussion

We explored several phase-dependent stability measures, to assess if they may adequately and consistently predict robustness of simple walking models. To do so, we devised five phase-dependent stability measures of compass walker models and investigated their correlations with gait robustness. We also calculated the commonly used local divergence exponent to test whether it may be outperformed by phase-dependent measures. Our analysis revealed that some phase-dependent stability measures could predict gait robustness better than the local divergence exponent for the simple point feet walker model, but failed to predict gait robustness for the circular feet walker. It is worth mentioning that in Bruijn *et al.* [33], the local divergence exponent was found to be well correlated with gait robustness of an arced feet walker with hip spring, but there the manipulated parameters were different (hip spring stiffness and foot radius) and much fewer configurations were tested in that study. Therefore, the correlations between stability measures and gait robustness also depend on which configuration parameters are varied.

We considered two different types of perturbations that took place at different phases during the gait cycle. We used a step-up/step-down positional perturbation that occurred at foot strike while push/pull and a force (pulse) perturbation that occurred around mid-stance phase. Even though the nature and timing of these two perturbations are completely different, there seems to be no qualitative difference in the correlations between stability measures and gait robustness quantified by these two different perturbations (figures 7 and 8). This may be explained by the fact that a step-down perturbation results in a larger stance leg velocity at the subsequent step [34], which is also the result of the push perturbations, and hence, the perturbation types may share similarities in their ultimate effects. In line with this, Chen *et al.* [35] applied acceleration disturbances at five different phases of a gait cycle for a planar-feet walker model. They found that the Pearson correlation coefficient of the maximum allowable perturbations added at different phases was higher than 0.65 for all pairs and greater than 0.8 for most pairs, suggesting that the measured maximum allowable perturbation (gait robustness) to be consistent along the gait cycle for simple walking models. Moreover, Bruijn *et al.* [33] reported similar correlations between local divergence exponent and gait robustness for a step-down perturbation and push/pull perturbation. Hence, when studying relationships between stability measures and robustness, using one robustness measure may suffice.

Our results showed that the phase-dependent stability measures depend on the state space form. This might be surprising, because the Lyapunov exponents are invariant to smooth coordinate transformation [4]. However, we note that all phase-dependent stability measures used here are *subspace* description of the original state space. They depend on the tangent vectors, which are not necessarily objective but may be coordinate- and norm-dependent. This may in fact be one of the reasons that different values

of phase-dependent stability measures are obtained when using different state variables or different scales of them. Similar arguments have been put forward by Nave *et al*. [32] and Sternad *et al*. [36]. Interestingly, we also found the local divergence rate (divergence rate without eliminating the phase shift dimension), which is not a subspace description, to be dependent on the state space form. We analysed this in appendix C.1 and found that due to the fact that the mass matrix is state-dependent, the state space transformation from angular velocities to Hamiltonian momenta is thus nonlinear. While we share the appeal in finding 'objective' stability measures that are state space form invariant (more precisely, the measures that remained unchanged under any translation or rotation of reference frame), we must admit that formulating them is far beyond the scope of the current paper. Yet, we dare to suggest an alternative, namely to find the 'best' state space representation, for instance, using Hamiltonian canonical form or action-angle coordinates [37]. However, this requires further studies that we leave for future work.

We considered most of the phase-dependent stability measures that are based on conventional local stability to have limited predictive value about gait robustness. Local stability quantifies a system's linear response under the proviso of infinitesimal perturbations. It, hence, does not necessarily reveal information about responses to larger perturbations. Nevertheless, predictions of gait robustness may be improved by looking at local stability throughout a gait cycle and calculating (novel) phase-dependent stability measures more appropriately. Interestingly, Ross *et al*. [38] probed the 'stability frontier' (i.e. the border between unstable and stable regions in state space) in biomechanical time series using local stability measures. Unfortunately, the outcome of this approach would, for our model, consist of high-dimensional dynamical boundaries that we consider difficult to interpret and requires further research. Moreover, their use for estimating the risk of falling remains to be validated.

Only the divergence of foot strike appeared to be reasonably well correlated with gait robustness of our two walker models. Both of the models have an instantaneous double stance phase. While this double stance phase is hardly comparable to that in humans, it is interesting to note that Ihlen *et al*. [12] found that during heel strike, older adults had impaired gait stability compared to younger adults. Also, older fallers have been reported to have higher gait instability and a higher phase-dependent entropy around 0 and 60% (heel strike and toe-off) of the gait cycle [19,21]. These findings do suggest that the stability of the foot strike could be of great importance in estimating/predicting gait robustness. A reason for that might be that (in our models) a dimension reduction occurs during foot strike, which constrains all post-impact perturbations in the state space into a two-dimensional plane. Apparently, a stability measure derived in that plane can contain valuable information about a model's robustness.

The divergence of foot strike does seem to be a promising measure, in particular when considering its consistent relation with gait robustness irrespective of state space form choices and numerical settings. Yet, note that we calculated this measure directly from the analytical expression of the walker model. When working with empirical data, a different approach is needed. For instance, one could use a method similar to Ihlen *et al*. [12] who calculated phase-dependent stability measures from experimental human data. Another interesting alternative, introduced by Nave *et al*. [32], is to calculate the trajectory divergence rate from vector fields obtained from experimental data. In this context, it is worth mentioning other potentially relevant variations of the trajectory-normal divergence rate, namely the *normal infinitesimal Lyapunov exponent* [39] that quantifies the instantaneous growth of perturbations transverse to the trajectory of the linearized dynamics, and finite-time Lyapunov exponents in the instantaneous limit [40], but both to our best knowledge have not yet been applied to time-series data. Finally, we would like to remark that there are of course also methods to calculate the entire Lyapunov spectrum in the presence of discontinuities, see, for instance, Müller [41], but supplemental numerical algorithms are far and few between and certainly not ready-for-use on empirical data. Moreover, since foot strike in human walking is not a discontinuity, it is still an open question whether such methods specifically designed for discontinuities should eventually be used.

## 4.1. Limitations of the current study

The phase-dependent stability measures used here are based on the idea that perturbations tangent to the period-one solution do not affect stability of the gait and will only result in a phase shift along the unperturbed trajectory. This idea has widely been used in controller design in engineering, for example, transverse linearization [42] and hybrid zero dynamics [43]. However, in human locomotion,

phase shift may play an important role, indeed. Stumbling perturbations applied at different phases can result in a phase reset to improve dynamic stability of gait [44]. Therefore, we also investigated the local divergence rate of the Jacobian $J(t)$ including both tangent and transverse local stability properties and found that the corresponding phase-dependent stability measures also did not strongly correlate with gait robustness (data not presented here but available via https://datadryad.org/stash/share/sXS8kl3SiaW1yteinh7-Tjuq5UP5iU5-pIu0lTbNGl8). Furthermore, we looked at the time-varying eigenvalue belonging to the eigenvector that is tangent to the period-one solution and found it to correlate only poorly with gait robustness.

Our phase-dependent stability measures have been derived based on coordinate transformation of the conventional local stability of time-invariant systems. Here, we would like to note that for time-varying systems, in general, the use of the Jacobian's eigenvalues to quantify local stability can be limited, because they may not account for the explicit time-dependent changes of the dynamics. However, our coordinate transformation can compensate for such time dependencies. In appendix D, we included an example of a two-dimensional time-varying system [45] illustrating this. Yet, a more formal mathematical analysis showing that coordinate transformation can improve local stability analysis is pending.

## 5. Conclusion

Phase-dependent stability measures have been used to characterize stability changes in compass walker models and in human gait. However, our simulations revealed that they do not correlate with gait robustness in a consistent manner. In particular, they may deviate strongly for different walker models and—arguably more important—they change with the choice of state space form. In our analysis, we encountered only a single phase-dependent stability measure, the divergence of the (instantaneous) double stance phase that displayed relatively good and coherent correlations with gait robustness. This almost linear relation, however, appears model dependent. Overall, it seems that even in simple compass walker models, phase-dependent stability measures and gait robustness are not one-to-one related, at least not the ones evaluated in the current study, which poses challenges for applying these measures to empirical data.

Data accessibility. Data and codes can be accessed from the Dryad Digital Repository: https://doi.org/10.5061/dryad.s4mw6m94r [46].

Authors' contributions. J.J. developed the theoretical framework, carried out the Matlab simulations, analysed the data and drafted the manuscript; D.K. verified and supervised the theoretical framework, encouraged J.J. to investigate different models and state space forms and provided critical revision of the manuscript; J.H.v.D. contributed to the methodology and data analysis, reviewed and revised the manuscript; A.D. contributed substantially to the theoretical framework, helped in writing, organizing and revising the manuscript; S.M.B. supervised the project, developed the theoretical framework, wrote and verified the code with J.J. and D.K., analysed the data with J.J., contributed substantially to the writing and revision of the manuscript. All authors gave final approval for publication.

Competing interests. We declare we have no competing interests.

Funding. S.M.B. and J.J. are funded by a VIDI grant no. (016.Vidi.178.014) from the Dutch Organization for Scientific Research (NWO).

Acknowledgements. The first author would like to thank Dr Shane Ross for his helpful comments on the differences between state space forms. We thank the Associate Editor Dr Manoj Srinivasan and three anonymous reviewers whose comments/suggestions helped improve and clarify this manuscript. We would like to thank Knoek van Soest for generously providing us the code for the (collision) dynamics of the curve feet walker model that he developed as course material for the Department of Human Movement Sciences, Vrije Universiteit, Amsterdam.

## Appendix A. Euler–Lagrange dynamics of the point feet walker

The gait cycle comprises a swing phase and an instantaneous double stance phase yielding one step. At double stance (foot strike), we assume a fully inelastic collision of the leading leg and account for the conservation of angular momentum. The swing phase equations of motion can be readily derived from

$$\frac{\mathrm{d}}{\mathrm{d}t}\frac{\partial \mathcal{L}}{\partial \dot{q}} - \frac{\partial \mathcal{L}}{\partial q} = 0 \ . \tag{A 1}$$

The state $q$ is given as $q = (\theta,\ \varphi)^T$ and $\mathcal{L}$ represents the Lagrangian. The corresponding equations of motion obey the form

$$M(\theta)\ddot{\theta} + N(\theta,\dot{\theta})\dot{\theta} + g(\theta) = \begin{bmatrix} 1 + 2\beta - 2\beta\cos\varphi & \beta(\cos\varphi - 1) \\ \beta(\cos\varphi - 1) & \beta \end{bmatrix} \begin{bmatrix} \ddot{\theta} \\ \ddot{\varphi} \end{bmatrix}$$

$$+ \begin{bmatrix} 2\beta\dot{\varphi}\sin\varphi & -\beta\dot{\varphi}\sin\varphi \\ -\beta\dot{\theta}\sin\varphi & 0 \end{bmatrix} \begin{bmatrix} \dot{\theta} \\ \dot{\varphi} \end{bmatrix} \tag{A2}$$

$$+ \begin{bmatrix} -(1+\beta)\sin(\theta - \gamma)Mgl + \beta\sin(\theta - \varphi - \gamma)Mgl \\ \beta Mgl\sin(\varphi + \gamma - \theta) \end{bmatrix} = 0\ ,$$

where $M(\theta)$ denotes the inertia matrix, $N(\theta,\dot{\theta})$ the Coriolis matrix and $g(\theta)$ contains all gravitational forces. By defining the state as $s = (\theta,\varphi,\dot{\theta},\dot{\varphi})^T$, one can rewrite the system (A2) in the Euler–Lagrange form

$$\dot{s} = f_{\mathrm{EL}}(s)\ . \tag{A3}$$

The foot strike occurs when the swing leg passes in front of the stance leg and hits the ground, that is, when $\cos\theta - \cos(\phi - \theta) = 0$ or simply $\phi - 2\theta = 0$ holds. To avoid foot scuffing, we add $\theta < -0.05$ as additional, necessary condition. Then, foot strike collision is instant, and can be described by a linear operator from the pre-state ($s^-$) to the post-state ($s^+$)

$$s^+ = \begin{bmatrix} \theta \\ \varphi \\ \dot{\theta} \\ \dot{\varphi} \end{bmatrix}^+ = \begin{bmatrix} 1 & -1 & 0 & 0 \\ 0 & -1 & 0 & 0 \\ 0 & 0 & \dfrac{\cos 2\theta}{1 + \beta\sin^2\varphi} & 0 \\ 0 & 0 & \dfrac{\cos 2\theta(1 - \cos 2\theta)}{1 + \beta\sin^2\varphi} & 0 \end{bmatrix} \begin{bmatrix} \theta \\ \varphi \\ \dot{\theta} \\ \dot{\varphi} \end{bmatrix}^- = G_{\mathrm{EL}}(s^-)\ . \tag{A4}$$

# Appendix B. Hamiltonian dynamics of the point feet walker

The swing phase dynamics can be given by four differential equations

$$\dot{s} = \begin{bmatrix} \dfrac{p_\theta + (1 - \cos\varphi)p_\varphi}{1 + \beta\sin^2\varphi} \\[2mm] \dfrac{p_\varphi + \beta(1 - \cos\varphi)(p_\theta + 2p_\varphi)}{\beta(1 + \beta\sin^2\varphi)} \\[2mm] -(1 + \beta)\sin(\gamma - \theta) + \beta\sin(\gamma - \theta + \varphi) \\[2mm] \dfrac{4\beta\cos\varphi\sin\varphi\left[p_\varphi^2\left(\dfrac{1}{\beta} - 2\cos\varphi + 2\right) + p_\theta^2 - p_\varphi p_\theta(2\cos\varphi - 2)\right]}{(2\beta\sin^2\varphi + 2)^2} \\[2mm] -\dfrac{2p_\varphi^2\sin\varphi + 2p_\theta p_\varphi\sin\varphi}{2\beta\sin^2\varphi + 2} - \beta\sin(\varphi + \gamma - \theta) \end{bmatrix} = f_H(s)\ , \tag{B1}$$

where the state is defined as $s = (\theta,\ \varphi,\ p_\theta,\ p_\varphi)^T$. We would like to note that this set of differential equations is different from those described in Norris et al. [11], which contains an error in the term for $p_\varphi$.

As before, the foot strike collision is an instantaneous transition, described by a linear operator from the pre-state ($s^-$) to the post-state ($s^+$) that now reads

$$s^+ = \begin{bmatrix} \theta \\ \varphi \\ p_\theta \\ p_\varphi \end{bmatrix}^+ = \begin{bmatrix} 1 & -1 & 0 & 0 \\ 0 & -1 & 0 & 0 \\ 0 & 0 & \dfrac{\cos\varphi}{1 + \beta\sin^2\varphi} & \dfrac{\cos\varphi(1 - \cos\varphi)}{1 + \beta\sin^2\varphi} \\ 0 & 0 & 0 & 0 \end{bmatrix} \begin{bmatrix} \theta \\ \varphi \\ p_\theta \\ p_\varphi \end{bmatrix}^- = G_H(s^-)\ . \tag{B2}$$

# Appendix C. Dependency on the state space form

## C.1. Local divergence rate

In Cartesian coordinates, the generalized Hamiltonian momenta coincide with mechanical ones, i.e. the product of the inertia matrix and angular velocity

$$p = \begin{bmatrix} p_\theta \\ p_\varphi \end{bmatrix} = M\dot{q} = \begin{bmatrix} 1 + 2\beta - 2\beta\cos\varphi & \beta(\cos\varphi - 1) \\ \beta(\cos\varphi - 1) & \beta \end{bmatrix} \begin{bmatrix} \dot{\theta} \\ \dot{\varphi} \end{bmatrix}. \tag{C1}$$

Taking the derivative of this equation gives

$$\begin{bmatrix} \dot{q} \\ \dot{p} \end{bmatrix} = \begin{bmatrix} 1 & 0 \\ \dot{M} & M \end{bmatrix} \begin{bmatrix} \dot{q} \\ \ddot{q} \end{bmatrix}. \tag{C2}$$

Let us write the Euler–Lagrange dynamics in state space form

$$\begin{bmatrix} \dot{q} \\ \ddot{q} \end{bmatrix} = f_{EL}(q,\dot{q}) = \begin{bmatrix} 0 & I \\ f_{EL}^{21}(q,\dot{q}) & f_{EL}^{22}(q,\dot{q}) \end{bmatrix} \begin{bmatrix} q \\ \dot{q} \end{bmatrix} = \begin{bmatrix} 0 & I \\ f_{EL}^{21}(q,\dot{q}) & f_{EL}^{22}(q,\dot{q}) \end{bmatrix} \begin{bmatrix} I & 0 \\ 0 & M^{-1} \end{bmatrix} \begin{bmatrix} q \\ p \end{bmatrix}, \tag{C3}$$

where $I$ denotes the $2 \times 2$ identity matrix and $0$ the $2 \times 2$ zero matrix and $f_{EL}^{21}(q,\dot{q})$ and $f_{EL}^{22}(q,\dot{q})$ are both $2 \times 2$ matrices. By combining (C 2) and (C 3), one can transform from the Euler–Lagrange state space into Hamiltonian form (see also appendix B)

$$\begin{bmatrix} \dot{q} \\ \dot{p} \end{bmatrix} = \begin{bmatrix} 1 & 0 \\ \dot{M} & M \end{bmatrix} \begin{bmatrix} 0 & I \\ f_{EL}^{21}(q,\dot{q}) & f_{EL}^{22}(q,\dot{q}) \end{bmatrix} \begin{bmatrix} I & 0 \\ 0 & M^{-1} \end{bmatrix} \begin{bmatrix} q \\ p \end{bmatrix} = f_H(q,p). \tag{C4}$$

Since the system is conservative, the local divergence rate of $f_H(q, p)$ vanishes [37], i.e. we have

$$\text{div } f_H = \text{trace} \left( \begin{bmatrix} 1 & 0 \\ \dot{M} & M \end{bmatrix} \begin{bmatrix} 0 & I \\ f_{EL}^{21}(q,\dot{q}) & f_{EL}^{22}(q,\dot{q}) \end{bmatrix} \begin{bmatrix} I & 0 \\ 0 & M^{-1} \end{bmatrix} \right) = 0. \tag{C5}$$

Simplifying (C 5) yields

$$\text{div } f_{EL} = \text{trace } f_{EL}^{22}(q,\dot{q}) = -\text{trace}(M^{-1}\dot{M}) = -\frac{\beta\sin 2\phi}{1 + \beta\sin^2\phi}\dot{\phi}, \tag{C6}$$

which implies that local divergence rate, quantified by the sum of eigenvalues of the Jacobian matrix, are different for two sets of state space forms in the point feet walker. For Hamiltonian form, local divergence rate equals zero; for the Euler–Lagrange form, local divergence rate equals $-(\beta\sin 2\phi/1 + \beta\sin^2\phi)\dot{\phi}$. This difference stems from the nonlinearity of the transformation $p = M\dot{q}$ as the inertia matrix $M$ is state-dependent.

## C.2. Phase-dependent stability measures

Figure 10 shows a period-one solution $[s_1(t), s_3(t), s_4(t)]$ during swing phase in both Hamiltonian canonical form $\left( \begin{bmatrix} \theta & p_\theta & p_\varphi \end{bmatrix}^T \right)$ and Euler–Lagrange form $\left( \begin{bmatrix} \theta & \dot{\theta} & \dot{\varphi} \end{bmatrix}^T \right)$. The final state of the period-one trajectory $p_\varphi$ in Hamiltonian canonical form is very close to zero. We found the relations between $p_\theta$ and $\dot{\theta}$ and between $p_\varphi$ and $\dot{\varphi}$ to be almost linear: $\dot{\theta} \approx 0.994 p_\theta$ and $\dot{\varphi} \approx 24.1032 p_\varphi$. That is, a transforming from Hamiltonian to Euler–Lagrange form may be approximate by keeping the scale of abcissa and rescaling ordinate by a factor of 24.

In figure 11, we illustrate the influence of coordinate transformation to the tangent and orthogonal vectors by sketching three sets of tangent/orthogonal vectors: one in Hamiltonian canonical form (purple), one in Euler–Lagrange form (green) and the other transformed from the Euler–Lagrange to Hamiltonian canonical form (orange). Note that the transformed tangent vector is still parallel to the tangent vector in Hamiltonian form, but the transformed orthogonal vector is no longer normal to the transformed tangent vector due to coordinate transformation. As a result, this type of coordinate transformation (scaling) makes our phase-dependent stability measures dependent on the state space form in which they are calculated. In other words, the derived phase-dependent stability measures are only specific to particular coordinate choice or state space form.

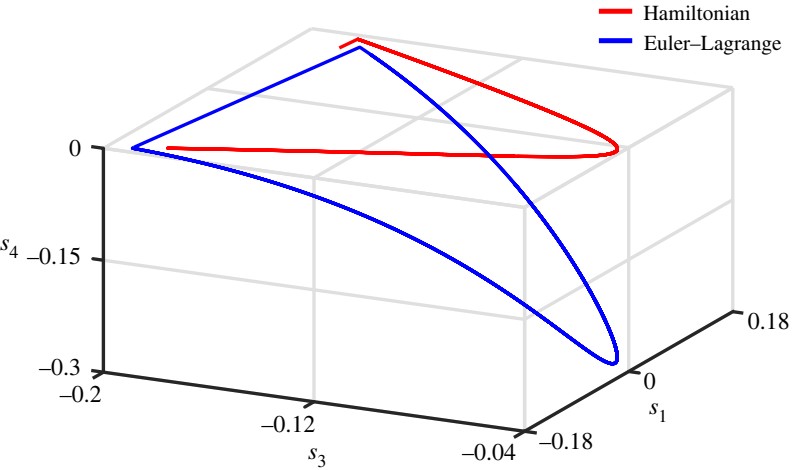

**Figure 10.** A period-one solution during swing phase in both Hamiltonian canonical form (red) and Euler–Lagrange form (blue).

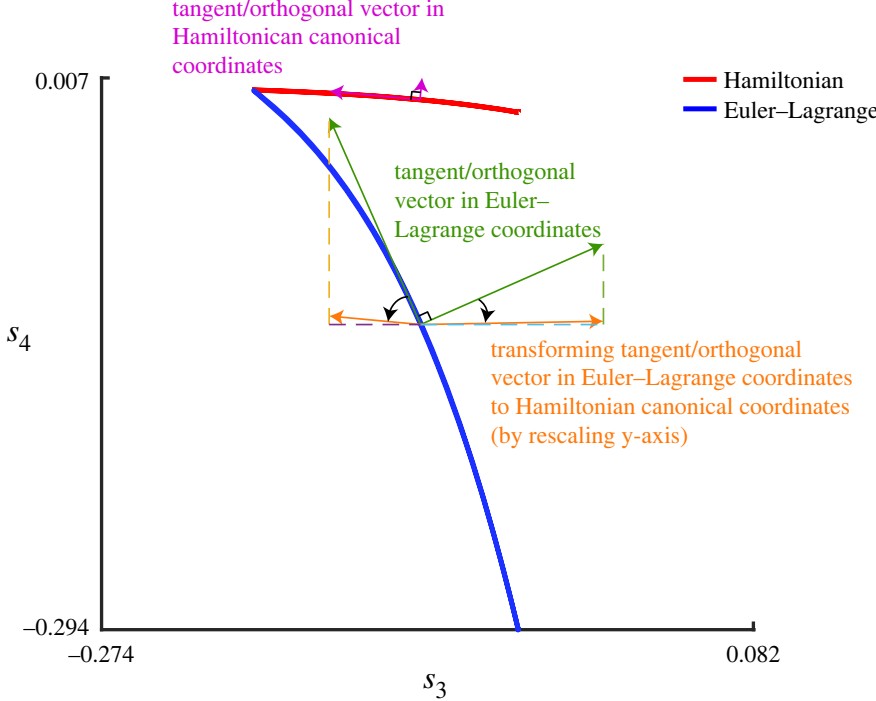

**Figure 11.** A period-one solution during swing phase in both Hamiltonian canonical form (red) and Euler–Lagrange form (blue). Three sets of tangent/orthogonal vectors are illustrated: in Hamiltonian canonical form (purple), in Euler–Lagrange form (green), transformed from the Euler–Lagrange to Hamiltonian canonical form (orange).

## Appendix D. A time-varying system example

Example: (adapted from Leonov & Kuznetsov [45])

Consider the linear system of non-autonomous differential equations

$$\dot{x} = A(t)x \ \text{ with } t \to x \colon \mathbb{R} \mapsto \mathbb{R}^2, \tag{D1}$$

with periodic coefficients

$$A(t) = \begin{pmatrix} 1 - 4\cos^2(2t) & 2 + 2\sin(4t) \\ -2 + 2\sin(4t) & 1 - 4\sin^2(2t) \end{pmatrix}. \tag{D2}$$

$A(t)$ defines a defective system with degenerated eigenvalues $\alpha_1 = \alpha_2 = -1$. That is, although $A(t)$, its real-valued eigenvalues do not depend on time and are negative, which indicates *local stability everywhere*.

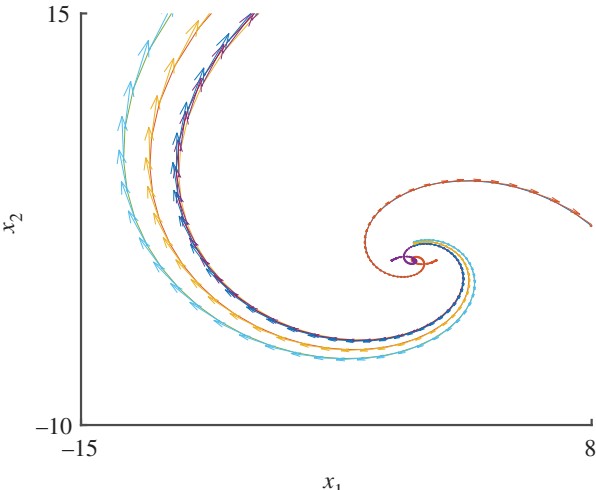

**Figure 12.** Trajectories of $x(t)$ for five different initial conditions; arrows indicating the vector field in their immediate vicinity. Note that only a small range of state values is shown.

On the other hand, the function

$$x(t) = \begin{pmatrix} e^t \sin(2t) \\ e^t \cos(2t) \end{pmatrix} \tag{D 3}$$

is an explicit solution of (D 1). One can readily see that the solution (D 3) is unbounded, i.e. $\lim_{t \to \infty} x_2(t) = \lim_{t \to \infty} e^{2t} \to \infty$, which implies *global instability* of the dynamics (D 1). And, since (D 1) is linear in $x$, it also suggests local instability of (D 3).

This example readily shows that a 'time-frozen' linearization of the Jacobian matrix may not suffice to determine a system's or its solution's stability, in particular, if systems, e.g. their coefficients, change as a function of time. By contrast, due to our coordinate transform, our approach does account for such time dependencies (through the time derivative of the rotation matrix; see equation (D 3) in our Methods section) rendering the transformed Jacobian an arguably better starting point for identifying local instability. To illustrate this, we simulated several trajectories of (D 1) over a time interval of $t = 0 \dots 20$, with five different initial conditions at $t = 0$

$$x(0) \in \left\{ \begin{pmatrix} 1 \\ 0 \end{pmatrix}, \begin{pmatrix} -1 \\ 0 \end{pmatrix}, \begin{pmatrix} 0 \\ 1.1 \end{pmatrix}, \begin{pmatrix} 0 \\ 0.9 \end{pmatrix}, \begin{pmatrix} 0 \\ 1 \end{pmatrix} \right\}$$

The corresponding trajectories are shown in figure 12 (only a small range of state values are shown). Subsequently, we applied the coordinate transform, which, interestingly, led to a new, time-invariant Jacobian matrix, $\bar{J} = \begin{bmatrix} -2.2 & -1.6 \\ -1.6 & 0.2 \end{bmatrix}$, with two different, real-valued eigenvalues $\alpha_1 = -3$ and $\alpha_2 = 1$. The latter is positive, suggesting local instability of the corresponding solutions, i.e. this approach can identify local instability of the time-varying system. Of course, this statement remains a mere conjecture but we consider a rigorous proof beyond the scope of our current paper.

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
