## [Peer Review File · Royal Society Open Science]

Review History

RSOS-201122.R0 (Original submission)

Review form: Reviewer 1 (SD Ross)

Is the manuscript scientifically sound in its present form?

Yes

Are the interpretations and conclusions justified by the results?

Yes

Is the language acceptable?

Yes

Do you have any ethical concerns with this paper?

No

Have you any concerns about statistical analyses in this paper?

No

Recommendation?

Accept with minor revision (please list in comments)

Comments to the Author(s)

This paper presents a useful study of phase-dependent gait stability measures using two standard simple walking models. The work may have wider relevance than the authors intend as it applies to other systems with limit cycles in moderate dimensions (although the systems considered are hybrid continuous-discrete systems, which is an interesting wrinkle).

The comparison with what the authors term 'gait robustness' seemed vague to this reviewer at first, but after reading further, I understood and using that as a 'gold standard' seems appropriate. The hope, I suppose, is to predict gait robustness from some measure of dynamic stability, obtainable from in situ measurements or laboratory experiments.

"A dependency of the choice of coordinates would severely limit the usefulness of any measure for fall prediction in humans." This got me thinking that perhaps all stability metrics are doomed, as this coordinate dependence applies to any dynamical analysis technique. Mathematically, due to the "Straightening out theorem" for vector fields (related to the Frobenius theorem, see Abraham, Marsden, Ratiu (1983)), one can always perform a transformation to a choice of coordinates where the vector field, away from an equilibrium point, is very 'simple' and won't reveal any interesting structure (e.g., ANY even-dimensional vector field, whether Hamiltonian or not, can locally be transformed into action-angle variables). That said, some coordinate choices are better than others, for the phenomena of interest. Hopefully this paper will help propagate the notion of coordinate dependence of stability measures, as it is not widely appreciated.

In the results section, around Table 1, it was not mentioned what the meaning of the Kendall rank correlation close to 0 means. Since model 2E had a value close to zero, this seems relevant. Perhaps the authors could pick a threshold epsilon and if the absolute value of the Kendall rank correlation is $< \epsilon$, those boxes in Table 1 could be colored gray, indicating that it isn't a significant negative or positive correlation.

The discussion is a useful summary and meditation on the results, especially in light of the aforementioned hope (of relating robustness to local stability). But perhaps local stability near a limit cycle will never be able to reveal the robustness the author's seek. By its nature, local stability just quantifies what happens to infinitesimal perturbations. For gait robustness, there is likely a stability 'frontier' that is crossed by a perturbation. Is it reasonable to hope for a 'signature' of this, perhaps distant, frontier in state space from the local dynamics around a stable limit cycle?

In any case, the discussion is interesting and opens up future avenues for work. The authors mention a 'normal infinitesimal Lyapunov exponent', and I wonder if they are aware of the recent work of Nolan et al (2020, Nonlinear Dyn.) on finite-time Lyapunov exponents in the instantaneous limit. There may be a way to leverage recent divergence measures with biomechanics time series measurements, as explored by Ross et al (2010, Chaos).

This reviewer appreciates that the authors provide open-source code and data.

Minor corrections:

Page 2, line 51 (abstract): measures

Page 3, line 21: "fall prediction in elderly using this metric have an accuracy of 80% at best". 80% seems very good, yet it's said as if this is not very good.

Page 5, line 18: where (instead of 'which')

Page 6, line 32 and elsewhere: the superscripts for the powers of 10 are appearing as subscripts

Page 6, line 50: delete "maximum absolute"; instead write "Floquet multiplier with modulus > 1 ".

Page 6-7, the discussion of gait robustness was confusing. Perhaps including a schematic figure of the step-up/step-down perturbation and resulting motion would help.

Page 19, line 30, instead of "absolute eigenvalue > 1 ", use "eigenvalue modulus > 1 ")

Page 15, line 34: "An" instead of "And"

Page 18, References: Many of the journal references are missing volume and page information.

Review form: Reviewer 2

Is the manuscript scientifically sound in its present form?

Yes

Are the interpretations and conclusions justified by the results?

No

Is the language acceptable?

Yes

Do you have any ethical concerns with this paper?

No

Have you any concerns about statistical analyses in this paper?

No

Recommendation?

Major revision is needed (please make suggestions in comments)

Comments to the Author(s)

This manuscript describes a model simulation study in which various measures of bipedal walking stability are compared to a measure of gait robustness to perturbations. This is an important topic, as it is presently unclear the extent to which existing stability measures are able to predict whether a given gait pattern is particularly susceptible to falls (a likely lay-person definition for "stability"). While the overall results are perhaps disappointing in terms of the link between stability and robustness, I think this work can be an important early step in this area. I have a few comments that I hope will help to improve clarity for readers.

Major comments

1. The importance of gait phase-dependence is in part justified by noting that perturbations applied during unstable gait phases may have larger effects than perturbations applied during stable gait phases (page 4, lines 7-8). But my impression is that the perturbations used to quantify the robustness of the walking model were all applied at the same point in the gait cycle (foot-strike, due to the model being a different configuration when making contact with the altered

ground surface). Could this approach have contributed to the finding that only a stability measure of foot-strike divergence consistently correlated with gait robustness?

2. I'm a little confused by the description of the calculated eigenvalues. On page 7, these are described as quantifying "the rate at which the system returns to (negative eigenvalue) or moves away from (positive eigenvalue) the period-one solution". My impression was that larger magnitude eigenvalues (whether positive or negative) would cause the system to move more quickly away from the limit cycle, so eigenvalues either greater than +1 or less than -1 would be considered unstable (as stated on page 10). Am I misunderstanding the authors' description?

Possibly related to this, I think that the description of the "trajectory-normal divergence rate" metric would benefit from more explanation. Based on this metric, it seems that a system with two eigenvalues of +1 and -1 at some point in the gait cycle would be defined to have the same "stability" as a system with two eigenvalues of 0 and 0. This seems counterintuitive to me. Similarly, could Figure 4 be explained in a little more detail? How many eigenvalues are plotted here? While I understand that they may be overlapping, it's not clear from the figure that the eigenvalues sum to the trajectory-normal divergence rate.

Minor comments

1. If possible given word limits, I think it would be helpful to readers to briefly define stability and robustness in the Abstract.

2. Was there a maximum of one period-one solution identified for each parameter combination? (see page 6, line 43) I seem to remember from the early McGeer simulations that some parameters produced multiple limit cycles – such as a short-period and long-period gait. Maybe these "extra" solutions were prevented by the other criteria used in this manuscript.

3. In Table 1, could you explicitly state what the models are?

4. This may just be a matter of semantics (and does not necessarily need to be addressed), but is foot-strike divergence truly a "phase-dependent stability metric", as implied on page 11 line 49? By looking at only one point in the gait cycle, this would appear to be fairly similar to previous stability metrics that I don't think were described as "phase-dependent".

Review form: Reviewer 3

Is the manuscript scientifically sound in its present form?

No

Are the interpretations and conclusions justified by the results?

No

Is the language acceptable?

Yes

Do you have any ethical concerns with this paper?

No

Have you any concerns about statistical analyses in this paper?

No

Recommendation?

Reject

Comments to the Author(s)

This study investigated whether different measures of phase-dependent local dynamic stability can predict gait robustness of different models of compass-walkers. The study found that for most of the measures the phase-dependent local dynamic stability was dependent on model type and on coordinate representations. The paper is well structured and well written. However, I have some major and general concerns with this study:

1. The use of phase-dependent local dynamic stability for assessment of gait instabilities and prediction of falls is a quite narrow field of research with a limited number of contributions. The authors state in the abstract "Current stability measures that estimate gait stability and robustness appear limited in predicting falls in older adults" for which I completely agree. Even though there are a paper (Ihlen et al., 2015) showing increased phase-dependent instabilities for fallers compared to non-fallers, there is close to no empirical evidence for the importance of these measurements in fall prediction. Thus, I do not see the importance of testing these measures on simple models of compass-walkers without more evidence in favor of their strong predictive value.
2. It is a paradox to me, why the authors do not include the measure utilized in Ihlen et al (2015) since this is (to the reviewer's knowledge) the only empirical evidence of phase-dependent local dynamic stability predicting falls in older persons. A major challenge of the definition used in Ihlen et al (2015) is that the phase-dependent stability refers to a percentage of the gait cycle and not to events like foot-strike and swing phase used in the authors study. Still, the inclusion of the measure would have increased the relevance of the paper for the development of new biometrics for fall prediction.
3. Even though the results of present study may shed some light on why phase-dependent stability is not a sensitive predictor of falls, there is probably numerous other factors, like the problem of noise and non-stationarities affecting the numerical estimation of these measures from data, that might be more interesting for the general audience of this journal.

In conclusion, I am struggling to see the clear scientific relevance of the present study in the important field of developing new biometrics for fall prediction. However, the study may be a more important contribution in analyzing the gait robustness of simple humanoid robots. Thus, I can not recommend this manuscript to be published in Royal Society Open Science in its current form.

Decision letter (RSOS-201122.R0)

Dear Dr Bruijn

The Editors assigned to your paper RSOS-201122 "The validation of new phase-dependent gait stability measures: a modeling approach" have now received comments from reviewers and would like you to revise the paper in accordance with the reviewer comments and any comments from the Editors. Please note this decision does not guarantee eventual acceptance.

Please submit your revised manuscript and required files (see below) no later than 21 days from today's (ie 02-Oct-2020) date. Note: the ScholarOne system will 'lock' if submission of the revision is attempted 21 or more days after the deadline. If you do not think you will be able to meet this deadline please contact the editorial office immediately.

on behalf of Dr Manoj Srinivasan (Associate Editor) and R. Kerry Rowe (Subject Editor)
openscience@royalsociety.org

Associate Editor Comments to Author (Dr Manoj Srinivasan):

The authors use the eigenvalues of the dependent jacobian to construct stability metrics. However, I would like to point out the following important and counter intuitive mathematical fact. Given a linear time varying ODE of the form

$dx/dt = J(t) x$. (same as authors' equation 2 with x instead of δ)

Having eigenvalues of $J(t)$ all have negative real parts for all time t (ie., 'always' converging toward $x = 0$ by the authors' reasoning) DOES NOT necessarily imply asymptotic stability of $x = 0$.

That is, the usual stability result and intuition when J is a constant does not extend to when J is time-varying.

See pages 23-24 of the following text for an illustrative example:

https://www.researchgate.net/profile/Fritz_Colonius/publication/242154014_2008_IMA_PI_Summer_Program_for_Graduate_Students_Linear_Algebra_and_Applications/links/540714650cf2c48563b2914a.pdf

I'd suggest that the authors re-frame their approach and results in the light of these mathematics. In particular, it would be good to check their correlations between the different stability/robustness metrics for the simple example above, if possible.

Reviewer comments to Author:

Reviewer: 1

Comments to the Author(s)

This paper presents a useful study of phase-dependent gait stability measures using two standard simple walking models. The work may have wider relevance than the authors intend as it applies to other systems with limit cycles in moderate dimensions (although the systems considered are hybrid continuous-discrete systems, which is an interesting wrinkle).

The comparison with what the authors term 'gait robustness' seemed vague to this reviewer at first, but after reading further, I understood and using that as a 'gold standard' seems appropriate. The hope, I suppose, is to predict gait robustness from some measure of dynamic stability, obtainable from in situ measurements or laboratory experiments.

"A dependency of the choice of coordinates would severely limit the usefulness of any measure for fall prediction in humans." This got me thinking that perhaps all stability metrics are doomed, as this coordinate dependence applies to any dynamical analysis technique. Mathematically, due to the "Straightening out theorem" for vector fields (related to the Frobenius theorem, see Abraham, Marsden, Ratiu (1983)), one can always perform a transformation to a choice of coordinates where the vector field, away from an equilibrium point, is very 'simple' and won't reveal any interesting structure (e.g., ANY even-dimensional vector field, whether Hamiltonian or not, can locally be transformed into action-angle variables). That said, some coordinate choices are better than others, for the phenomena of interest. Hopefully this paper will help propagate the notion of coordinate dependence of stability measures, as it is not widely appreciated.

In the results section, around Table 1, it was not mentioned what the meaning of the Kendall rank correlation close to 0 means. Since model 2E had a value close to zero, this seems relevant. Perhaps the authors could pick a threshold epsilon and if the absolute value of the Kendall rank correlation is $< \epsilon$, those boxes in Table 1 could be colored gray, indicating that it isn't a significant negative or positive correlation.

The discussion is a useful summary and meditation on the results, especially in light of the aforementioned hope (of relating robustness to local stability). But perhaps local stability near a limit cycle will never be able to reveal the robustness the author's seek. By its nature, local stability just quantifies what happens to infinitesimal perturbations. For gait robustness, there is likely a stability 'frontier' that is crossed by a perturbation. Is it reasonable to hope for a 'signature' of this, perhaps distant, frontier in state space from the local dynamics around a stable limit cycle?

In any case, the discussion is interesting and opens up future avenues for work. The authors mention a 'normal infinitesimal Lyapunov exponent', and I wonder if they are aware of the recent work of Nolan et al (2020, Nonlinear Dyn.) on finite-time Lyapunov exponents in the instantaneous limit. There may be a way to leverage recent divergence measures with biomechanics time series measurements, as explored by Ross et al (2010, Chaos).

This reviewer appreciates that the authors provide open-source code and data.

Minor corrections:

Page 2, line 51 (abstract): measures

Page 3, line 21: “fall prediction in elderly using this metric have an accuracy of 80% at best”. 80% seems very good, yet it’s said as if this is not very good.

Page 5, line 18: where (instead of ‘which’)

Page 6, line 32 and elsewhere: the superscripts for the powers of 10 are appearing as subscripts

Page 6, line 50: delete “maximum absolute”; instead write “Floquet multiplier with modulus > 1 ”.

Page 6-7, the discussion of gait robustness was confusing. Perhaps including a schematic figure of the step-up/step-down perturbation and resulting motion would help.

Page 19, line 30, instead of “absolute eigenvalue > 1 ”, use “eigenvalue modulus > 1 ”)

Page 15, line 34: “An” instead of “And”

Page 18, References: Many of the journal references are missing volume and page information.

Reviewer: 2

Comments to the Author(s)

This manuscript describes a model simulation study in which various measures of bipedal walking stability are compared to a measure of gait robustness to perturbations. This is an important topic, as it is presently unclear the extent to which existing stability measures are able to predict whether a given gait pattern is particularly susceptible to falls (a likely lay-person definition for “stability”). While the overall results are perhaps disappointing in terms of the link between stability and robustness, I think this work can be an important early step in this area. I have a few comments that I hope will help to improve clarity for readers.

Major comments

1. The importance of gait phase-dependence is in part justified by noting that perturbations applied during unstable gait phases may have larger effects than perturbations applied during stable gait phases (page 4, lines 7-8). But my impression is that the perturbations used to quantify the robustness of the walking model were all applied at the same point in the gait cycle (foot-strike, due to the model being a different configuration when making contact with the altered ground surface). Could this approach have contributed to the finding that only a stability measure of foot-strike divergence consistently correlated with gait robustness?

2. I’m a little confused by the description of the calculated eigenvalues. On page 7, these are described as quantifying “the rate at which the system returns to (negative eigenvalue) or moves away from (positive eigenvalue) the period-one solution”. My impression was that larger magnitude eigenvalues (whether positive or negative) would cause the system to move more quickly away from the limit cycle, so eigenvalues either greater than +1 or less than -1 would be considered unstable (as stated on page 10). Am I misunderstanding the authors’ description?

Possibly related to this, I think that the description of the “trajectory-normal divergence rate” metric would benefit from more explanation. Based on this metric, it seems that a system with two eigenvalues of +1 and -1 at some point in the gait cycle would be defined to have the same “stability” as a system with two eigenvalues of 0 and 0. This seems counterintuitive to me. Similarly, could Figure 4 be explained in a little more detail? How many eigenvalues are plotted here? While I understand that they may be overlapping, it’s not clear from the figure that the eigenvalues sum to the trajectory-normal divergence rate.

Minor comments

1. If possible given word limits, I think it would be helpful to readers to briefly define stability and robustness in the Abstract.
2. Was there a maximum of one period-one solution identified for each parameter combination? (see page 6, line 43) I seem to remember from the early McGeer simulations that some parameters produced multiple limit cycles – such as a short-period and long-period gait. Maybe these “extra” solutions were prevented by the other criteria used in this manuscript.
3. In Table 1, could you explicitly state what the models are?
4. This may just be a matter of semantics (and does not necessarily need to be addressed), but is foot-strike divergence truly a “phase-dependent stability metric”, as implied on page 11 line 49? By looking at only one point in the gait cycle, this would appear to be fairly similar to previous stability metrics that I don’t think were described as “phase-dependent”.

Reviewer: 3

Comments to the Author(s)

This study investigated whether different measures of phase-dependent local dynamic stability can predict gait robustness of different models of compass-walkers. The study found that for most of the measures the phase-dependent local dynamic stability was dependent on model type and on coordinate representations. The paper is well structured and well written. However, I have some major and general concerns with this study:

1. The use of phase-dependent local dynamic stability for assessment of gait instabilities and prediction of falls is a quite narrow field of research with a limited number of contributions. The authors state in the abstract “Current stability measures that estimate gait stability and robustness appear limited in predicting falls in older adults” for which I completely agree. Even though there are a paper (Ihlen et al., 2015) showing increased phase-dependent instabilities for fallers compared to non-fallers, there is close to no empirical evidence for the importance of these measurements in fall prediction. Thus, I do not see the importance of testing these measures on simple models of compass-walkers without more evidence in favor of their strong predictive value.
2. It is a paradox to me, why the authors do not include the measure utilized in Ihlen et al (2015) since this is (to the reviewer’s knowledge) the only empirical evidence of phase-dependent local dynamic stability predicting falls in older persons. A major challenge of the definition used in Ihlen et al (2015) is that the phase-dependent stability refers to a percentage of the gait cycle and not to events like foot-strike and swing phase used in the authors study. Still, the inclusion of the measure would have increased the relevance of the paper for the development of new biometrics for fall prediction.

3. Even though the results of present study may shed some light on why phase-dependent stability is not a sensitive predictor of falls, there is probably numerous other factors, like the problem of noise and non-stationarities affecting the numerical estimation of these measures from data, that might be more interesting for the general audience of this journal.

In conclusion, I am struggling to see the clear scientific relevance of the present study in the important field of developing new biometrics for fall prediction. However, the study may be a more important contribution in analyzing the gait robustness of simple humanoid robots. Thus, I can not recommend this manuscript to be published in Royal Society Open Science in its current form.

===PREPARING YOUR MANUSCRIPT===

===PREPARING YOUR REVISION IN SCHOLARONE===

Author's Response to Decision Letter for (RSOS-201122.R0)

See Appendix A.

RSOS-201122.R1 (Revision)

Review form: Reviewer 1

Is the manuscript scientifically sound in its present form?

Yes

Are the interpretations and conclusions justified by the results?

Yes

Is the language acceptable?

Yes

Do you have any ethical concerns with this paper?

No

Have you any concerns about statistical analyses in this paper?

No

Recommendation?

Accept as is

Comments to the Author(s)

All of my concerns have been adequately addressed.

Review form: Reviewer 2

Is the manuscript scientifically sound in its present form?

Yes

Are the interpretations and conclusions justified by the results?

Yes

Is the language acceptable?

Yes

Do you have any ethical concerns with this paper?

No

Have you any concerns about statistical analyses in this paper?

No

Recommendation?

Accept as is

Comments to the Author(s)

The authors have clearly addressed my previous comments, with appropriate explanatory additions to the manuscript.

Decision letter (RSOS-201122.R1)

Dear Dr Bruijn,

It is a pleasure to accept your manuscript entitled "The validation of new phase-dependent gait stability measures: a modeling approach" in its current form for publication in Royal Society Open Science.

on behalf of Dr Manoj Srinivasan (Associate Editor) and R. Kerry Rowe (Subject Editor)
openscience@royalsociety.org

Reviewer comments to Author:
Reviewer: 2

Comments to the Author(s)
The authors have clearly addressed my previous comments, with appropriate explanatory additions to the manuscript.

Reviewer: 1
Comments to the Author(s)

All of my concerns have been adequately addressed.

Appendix A

Title: The validation of new phase-dependent gait stability measures: a modeling approach

Associate editor:

The authors use the eigenvalues of the dependent jacobian to construct stability metrics. However, I would like to point out the following important and counter intuitive mathematical fact. Given a linear time varying ODE of the form $dx/dt=J(t)x$ (same as authors' equation 2 with x instead of δ).

Having eigenvalues of $J(t)$ all have negative real parts for all time t (i.e., 'always' converging toward $x = 0$ by the authors' reasoning) DOES NOT necessarily imply asymptotic stability of $x=0$.

That is, the usual stability result and intuition when J is a constant does not extend to when J is time-varying.

See pages 23-24 of the following text for an illustrative example:

https://www.researchgate.net/profile/Fritz_Colonius/publication/242154014_2008_IMA_PI_Summer_Program_for_Graduate_Students_Linear_Algebra_and_Applications/links/540714650cf2c48563b2914a.pdf

I'd suggest that the authors re-frame their approach and results in the light of these mathematics. In particular, it would be good to check their correlations between the different stability/robustness metrics for the simple example above, if possible.

>> Thank you for pointing this out. We adopted your suggestion, which definitely helped improving the manuscript. In particular, we discussed in Appendix D the example you mentioned, which, we have learned, in fact dates back to Bylov et al., (1966) and found to be well summarized in Leonov & Kuznetsov, (2007, International Journal of Bifurcation and Chaos). Before sketching this discussion, however, please allow us to note that Eq.2 that you mention in your comment contains a coordinate transform, while Eq.1 represents an ODE. In the latter, we used the notion $J = J(t)$, which might have been misleading because our two models are 'time-invariant': J does only implicitly depend on time via the point $x(t)$ at which it is evaluated.

We agree that it is important to also discuss the dynamics with time-dependent coefficients in the light of the phase-dependent stability measures. You are right that in such cases, the real parts of the Jacobian's eigenvalues do not necessarily provide information about the stability of an explicit solution. When properly done, the (maximum) Lyapunov exponents used as measure of this local stability are defined in the time-asymptotic limit, which does not exist in the example that you provided. Hence, this approach fails. However, our phase-dependent stability measure successfully indicates local (in)stability. In our understanding, the validity of our approach stems from the coordinate transform in Eq.2 and Eq.3, which does account for time-dependent changes of the dynamics itself (through the time derivative of the rotational matrix U) rather than evaluating the local Jacobian that contains only the partial derivatives of the system matrix w.r.t. the states). As it stands, this is of course a (narrative) conjecture and a proper proof deserves more attention, but we feel that this is beyond the scope of the current paper. Yet, as said, we have used the very example that you mentioned to show that our stability measure is indeed able to adequately describe its stability properties;

this analysis is now in the new Appendix D. We also added a paragraph to describe the potential use of phase-dependent stability measures in dynamical systems with time-dependent coefficients since we believe that this may broaden the potential impact of our work and phase-dependent stability measures in general.

Example: (adapted from Colonius, 2018, IMA PI Summer Program for Graduate Students, Linear Algebra and Applications, Example 6.1; see also Leonov & Kuznetsov, 2007)

Consider the linear system of non-autonomous differential equations

$$\dot{x} = A(t)x \quad \text{with } t \rightarrow x: \mathbb{R} \mapsto \mathbb{R}^2 \quad (1)$$

with periodic coefficients

$$A(t) = \begin{pmatrix} 1 - 4 \cos^2(2t) & 2 + 2 \sin(4t) \\ -2 + 2 \sin(4t) & 1 - 4 \sin^2(2t) \end{pmatrix}. \quad (2)$$

$A(t)$ defines a defective system with degenerated eigenvalues $\alpha_1 = \alpha_2 = -1$. That is, although $A=A(t)$, its real-valued eigenvalues do not depend on time and are negative, which indicates *local stability everywhere*. On the other hand, the function

$$x(t) = \begin{pmatrix} e^t \sin(2t) \\ e^t \cos(2t) \end{pmatrix} \quad (3)$$

is an explicit solution of (1). One can readily see that the solution (3) is unbounded, i.e. $\lim_{t \rightarrow \infty} x^2(t) = \lim_{t \rightarrow \infty} e^{2t} \rightarrow \infty$, which implies *global instability* of the dynamics (1). And, since (1) is linear in x , it also suggests local instability of (3).¹

This example readily shows that a “time-frozen” linearization of the Jacobian matrix may not suffice to determine a system’s or its solution’s stability, in particular, if systems, e.g., their coefficients, change as a function of time. By contrast, due to our coordinate transform, our approach does account for such time dependencies (through the time derivative of the rotation matrix; see Equation 3 in our Methods section) rendering the transformed Jacobian an arguably better starting point for identifying local instability. To illustrate this, we simulated several trajectories of (1) over a time interval of $t = 0 \dots 20$, with five different initial conditions at $t = 0$:

$$x(0) \in \left\{ \begin{pmatrix} 1 \\ 0 \end{pmatrix}, \begin{pmatrix} -1 \\ 0 \end{pmatrix}, \begin{pmatrix} 0 \\ 1.1 \end{pmatrix}, \begin{pmatrix} 0 \\ 0.9 \end{pmatrix}, \begin{pmatrix} 0 \\ 1 \end{pmatrix} \right\}$$

The corresponding trajectories are shown in Figure 1 of this rebuttal. Subsequently, we applied the coordinate transform, which, interestingly, led to a new, time-invariant Jacobian matrix, $\bar{J} = \begin{bmatrix} -2.2 & -1.6 \\ -1.6 & 0.2 \end{bmatrix}$, with two different, real-valued eigenvalues $\alpha_1 = -3$ and $\alpha_2 = 1$. The latter is positive, suggesting local instability of the corresponding solutions, i.e. this approach can identify local instability of the time-varying system. Of course, this statement remains a mere conjecture but we consider a rigorous proof beyond the scope of our current paper (and of this response).

As said, next to Appendix D we also now express these ideas in the Discussion of our manuscript:

¹ This ‘dilemma’ does not contradict Lyapunov’s fundamental stability theorem as strictly but rather show that if (one of) its assumptions are violated, the outcomes can readily lead to false conclusion – here the asymptotic limit of the (singular values of the) fundamental matrix does not exist.

Figure 1: Trajectories of $x(t)$ for five different initial conditions; arrows indicating the vector field in their immediate vicinity. Note that only a small range of state values is shown.

Page 18, line 15 to 22: “Our phase-dependent stability measures have been derived based on coordinate transformation of the conventional local stability of time-invariant systems. Here, we would like to note that for time-varying systems, in general, the use of the Jacobian’s eigenvalues to quantify local stability can be limited, because they may not account for the explicit time-dependent changes of the dynamics. However, our coordinate transformation can compensate for such time-dependencies. In Appendix D, we included an example of a two-dimensional time-varying system (Leonov & Kuznetsov, 2007) illustrating this. Yet, a more formal mathematical analysis showing that coordinate transformation can improve local stability analysis is.”

Reviewer 1:

This paper presents a useful study of phase-dependent gait stability measures using two standard simple walking models. The work may have wider relevance than the authors intend as it applies to other systems with limit cycles in moderate dimensions (although the systems considered are hybrid continuous-discrete systems, which is an interesting wrinkle).

The comparison with what the authors term ‘gait robustness’ seemed vague to this reviewer at first, but after reading further, I understood and using that as a ‘gold standard’ seems appropriate. The hope, I suppose, is to predict gait robustness from some measure of dynamic stability, obtainable from in situ measurements or laboratory experiments.

“A dependency of the choice of coordinates would severely limit the usefulness of any measure for fall prediction in humans.” This got me thinking that perhaps all stability metrics are doomed, as this coordinate dependence applies to any dynamical analysis technique.

Mathematically, due to the “Straightening out theorem” for vector fields (related to the Frobenius theorem, see Abraham, Marsden, Ratiu (1983)), one can always perform a transformation to a choice of coordinates where the vector field, away from an equilibrium point, is very ‘simple’ and won’t reveal any interesting structure (e.g., ANY even-dimensional vector field, whether Hamiltonian or not, can locally be transformed into action-angle variables). That said, some coordinate choices are better than others, for the phenomena of interest. Hopefully this paper will help propagate the notion of coordinate dependence of stability measures, as it is not widely appreciated.

>> Thank you for the encouraging comments. As you can see in our point-by-point reply below, we adopted most of your suggestions as this clearly improved the legibility of our manuscript.

In the results section, around Table 1, it was not mentioned what the meaning of the Kendall rank correlation close to 0 means. Since model 2E had a value close to zero, this seems relevant. Perhaps the authors could pick a threshold epsilon and if the absolute value of the Kendall rank correlation is $< \epsilon$, those boxes in Table 1 could be colored gray, indicating that it isn’t a significant negative or positive correlation.

>> We entirely agree and, upon revision, highlighted the implications of Kendall’s rank correlation by displaying correlation coefficients >0.7 (or <-0.7) in bold font to indicate strong correlations. All of these strong correlations are statistically significant ($p < 0.001$). Note, however, that a “strong and significant” correlation does not necessarily mean that this measure is a good predictor for gait robustness. Consider, for instance, our new Figure 6E. A good predictor of gait robustness should be able to predict a limited range of robustness given a particular measure value. We added the following text in the Results section:

Page 11, line 23 to 29: “A coefficient value of 1 (-1) satisfies the monotonic relation that when one variable increases, the other variable increases (decreases); a coefficient value of 0 implies the absence of any association between the two variables. We considered Kendall rank correlation coefficients larger than 0.7 (or smaller than -0.7) to be indicative for strong correlations. For truly promising stability measures predicting the robustness of a human gait, however, we expect the correlation coefficients to be larger than 0.9 (or smaller than -0.9).”

The discussion is a useful summary and meditation on the results, especially in light of the aforementioned hope (of relating robustness to local stability). But perhaps local stability near a limit cycle will never be able to reveal the robustness the author’s seek. By its nature, local stability just quantifies what happens to infinitesimal perturbations. For gait robustness, there is likely a stability ‘frontier’ that is crossed by a perturbation. Is it reasonable to hope for a ‘signature’ of this, perhaps distant, frontier in state space from the local dynamics around a stable limit cycle?

>> We share your considerations about local stability (and mentioned them in our Methods section). To be honest, we do not know whether local stability has the potential to predict robustness for complex nonlinear systems like human gait. Based on recent literature, we had intuitive questions like: “is a system with locally very large eigenvalues

less robust than a system with eigenvalues just above 0?” and: “is a system that has positive eigenvalues along a substantial part of the trajectory less robust than a system with limited positive eigenvalues along the trajectory?” We therefore formulated our measures that rely on local stability. As said, they arguably do not probe the stability frontier like the method you mentioned does (Ross et al., 2010, Chaos). We did implement Ross’s algorithm in early stages of the project but since our walker models are four-dimensional, the outcomes of this algorithm (four-dimensional boundary curves) are difficult to visualize and too complicated (at least for us) to get one measure (to predict robustness) from it. We have now added an extra paragraph in the Discussion section to describe this method and the limitation of predicting robustness from conventional local stability measures.

Page 16, line 26 to page 17, line 3: “We considered most of the phase-dependent stability measures that are based on conventional local stability to have limited predictive value about gait robustness. Local stability quantifies a system’s linear response under the proviso of infinitesimal perturbations. It, hence, does not necessarily reveal information about responses to larger perturbations. Nevertheless, predictions of gait robustness may be improved by looking at local stability throughout a gait cycle and calculating (novel) phase-dependent stability measures more appropriately. Interestingly, Ross et al., (2010) probed the “stability frontier” (the border between unstable and stable regions in state space) in biomechanical time series. Unfortunately, the outcome of this approach would, for our model, consist of high-dimensional dynamical boundaries that we consider difficult to interpret and requires further research. Moreover, their use for estimating the risk of falling remains to be validated.”

In any case, the discussion is interesting and opens up future avenues for work. The authors mention a ‘normal infinitesimal Lyapunov exponent’, and I wonder if they are aware of the recent work of Nolan et al. (2020, Nonlinear Dyn.) on finite-time Lyapunov exponents in the instantaneous limit. There may be a way to leverage recent divergence measures with biomechanics time series measurements, as explored by Ross et al (2010, Chaos).

>> We followed the work by Nave et al. (2019, Nonlinear Dyn.) but missed the more recent one by Nolan et al. (2020, Nonlinear Dyn.). Thank you very much for pointing at it. We have incorporated this interesting article as follows:

Page 17, line 23 to 27: “...namely the normal infinitesimal Lyapunov exponent (Haller & Sapsis, 2010) that quantifies the instantaneous growth of perturbations transverse to the trajectory of the linearized dynamics, and finite-time Lyapunov exponents in the instantaneous limit (Nolan et al., 2020), but both to our best knowledge have not yet been applied to time series data.”

This reviewer appreciates that the authors provide open-source code and data.

>> Thank you. We believe this is the future of science as it allows for both replication and validation, and also prevents future studies from having to develop everything from scratch.

Minor corrections:

1) Page 2, line 51 (abstract): measures.

>> We changed “metrics” to “measures” throughout the manuscript.

2) Page 3, line 21: “fall prediction in elderly using this metric have an accuracy of 80% at best”. 80% seems very good, yet it’s said as if this is not very good.

>> We agree that when predicting whether a person is prone to fall or not, the 80% accuracy appears very good. Yet, such a prediction may still lead to a large number of both false positives (predicted fallers that are not prone to fall) and false negatives (predicted non-fallers that actually are prone to fall). That is, prediction accuracy ought to be higher. We rephrased the last few sentences of the first paragraph to strengthen this point:

Page 2, Line 11 to 15: “..., with accuracy of 80% at best (Van Schooten et al., 2016). Although 80% accuracy is a fair achievement in view of the complexity of the dynamics involved in human walking, it may still lead to a large number of false positives (predicted fallers that are not prone to fall) and – arguably worse – false negatives (predicted non-fallers that actually are prone to fall) in the elderly population. Therefore, a higher accuracy is of great importance.

3) Page 5, line 18: where (instead of ‘which’).

>> Done.

4) Page 6, line 32 and elsewhere: the superscripts for the powers of 10 are appearing as subscripts.

>> Done.

5) Page 6, line 50: delete “maximum absolute”; instead write “Floquet multiplier with modulus > 1 ”.

>> Done.

6) Page 6-7, the discussion of gait robustness was confusing. Perhaps including a schematic figure of the step-up/step-down perturbation and resulting motion would help.

>> As suggested, we added a figure to illustrate the step-up and step-down perturbation (see new Figure 3).

7) Page 10, line 30, instead of “absolute eigenvalue > 1 ”, use “eigenvalue modulus > 1 ”.

>> Done.

8) Page 15, line 34: “An” instead of “And”.

>> Done.

9) Page 18, References: Many of the journal references are missing volume and page information.

>> We apologize for these errors; they have been corrected.

Reviewer 2:

This manuscript describes a model simulation study in which various measures of bipedal walking stability are compared to a measure of gait robustness to perturbations. This is an important topic, as it is presently unclear the extent to which existing stability measures are able to predict whether a given gait pattern is particularly susceptible to falls (a likely lay-person definition for “stability”). While the overall results are perhaps disappointing in terms of the link between stability and robustness, I think this work can be an important early step in this area. I have a few comments that I hope will help to improve clarity for readers.

>> Thank you very much for the detailed comments. We revised the manuscript following your suggestions point by point.

Major comments:

1) The importance of gait phase-dependence is in part justified by noting that perturbations applied during unstable gait phases may have larger effects than perturbations applied during stable gait phases (page 4, lines 7-8). But my impression is that the perturbations used to quantify the robustness of the walking model were all applied at the same point in the gait cycle (foot-strike, due to the model being a different configuration when making contact with the altered ground surface). Could this approach have contributed to the finding that only a stability measure of foot-strike divergence consistently correlated with gait robustness?

>> You raise a very interesting point and we decided to thoroughly analyze this. The step-up/step-down perturbation is a typical perturbation, which by definition happens at heel strike. Although the effects of this perturbation can, and typically do, persist throughout gait cycle, it might be that the nature and timing of the perturbation influences the correlations between the here-investigated measures and gait robustness. To further investigate this issue, we simulated another type of perturbation on the circular feet walker in a different phase of the gait cycle: a one-time constant push (or pull) perturbation, starting when the hip angle is zero (close to mid-stance phase) that lasted for 0.1s. Then, we quantify gait robustness as the sum of the maximum allowable push and pull perturbation without falling. The relations between different stability measures and gait robustness are shown in Figure 2 below.

Figure 2. Relation between 1+5 stability measures and gait robustness (quantified by the sum of maximum allowable push and pull perturbation) for the circular feet walker. Panel A represents the local divergence exponent, panels B to F represent phase-dependent stability measures i to v respectively. Every data point represents a parameter combination of the model.

Based on these plots, the Kendall Rank correlations coefficients looks as follows:

Table 1. Kendall rank correlation coefficients of stability measures with gait robustness for the different models, different state space forms and different perturbation types. Green shading indicates negative correlations (higher stability measure value corresponding to lower robustness, as theoretically expected) and grey shading indicates positive correlations. We considered Kendall rank correlation coefficients larger than the 0.7 (or smaller than -0.7) to be strong correlations and displayed them in bold font in the table. All of these strong correlations are statistically significant ($p < 0.001$).

model	perturbation	local div. exponent	max. div. rate (i)	min. div. rate (ii)	div. swing ph. (iii)	div. foot strike (iv)	div. gait cycle (v)
1_{Euler}	step-up/down	-0.79	-0.47	-0.46	-0.97	-0.96	-0.25
$1_{\text{Hamiltonian}}$	step-up/down	-0.54	-0.93	0.61	-0.93	-0.96	-0.25
2_{Euler}	step-up/down	-0.01	0.05	0.87	0.74	-0.80	0.57
2_{Euler}	push/pull	0.13	-0.003	0.84	0.67	-0.69	0.62

From this new table one can see that replacing the step-up and step-down perturbation with push and pull perturbation, applied at a different phase, will not alter our general conclusions, at least not qualitatively. The results are very similar to the outcome of the step-up step-down perturbation (c.f. new Figure 7 in the manuscript). However, the measure of divergence of foot strike appears weaker correlated with gait robustness (from -0.797 to -0.688). In the Methods, Results and Discussion section of our manuscript, we added the following text to describe these new findings:

Page 6, line 12 to 17, Methods section: “To generalize gait robustness in the light of another type of perturbation, we also applied a one-time constant push or pull. We performed this type of perturbation only for the circular feet walker. The push or pull was applied during the first step for 0.1 s, starting at the moment the hip angle was zero.

Gait robustness was then quantified as the sum of the maximum allowable push and pull perturbation or the sum of maximum step-up and step-down perturbation without falling.”

Page 13, line 10 to 13, Results section: “Interestingly, applying different types of perturbations to quantify gait robustness (maximum step-up/step-down perturbation in Fig. 7 and maximum push/pull perturbation in Fig. 8) did not alter the general relations between stability measures and gait robustness, at least not qualitatively.”

Page 15, line 24 to page 16, line 7, Discussion section: “We considered two different types of perturbations that took place at different phases during the gait cycle. We used a step-up/step-down positional perturbation that occurred at foot strike while push/pull and a force (pulse) perturbation that occurred around mid-stance phase. Even though the nature and timing of these two perturbations are completely different, there seems to be no qualitative difference in the correlations between stability measures and gait robustness quantified by these two different perturbations (Fig. 7 and 8). This may be explained by the fact that a step-down perturbation results in a larger stance leg velocity at the subsequent step (Wisse et al., 2005), which is also the result of the push perturbations, and hence, the perturbation types may share similarities in their ultimate effects. In line with this, Chen et al., (2020) applied acceleration disturbances at five different phases of a gait cycle for a planar-feet walker model. They found that the Pearson correlation coefficient of the maximum allowable perturbations added at different phases were higher than 0.65 for all pairs and greater than 0.8 for most pairs, suggesting that the measured maximum allowable perturbation (gait robustness) to be consistent along the gait cycle for simple walking models. Moreover, Bruijn et al., (2012) reported similar correlations between local divergence exponent and gait robustness for a step-down perturbation and push/pull perturbation. Hence, when studying relationships between stability measures and robustness, using one robustness measure may suffice.”

2) I’m a little confused by the description of the calculated eigenvalues. On page 7, these are described as quantifying “the rate at which the system returns to (negative eigenvalue) or moves away from (positive eigenvalue) the period-one solution”. My impression was that larger magnitude eigenvalues (whether positive or negative) would cause the system to move more quickly away from the limit cycle, so eigenvalues either greater than +1 or less than -1 would be considered unstable (as stated on page 10). Am I misunderstanding the authors’ description?

>> Actually, both are true. For a discrete system ($x(i + 1) = A \cdot x(i)$ a difference equation), e.g., a foot strike or a full gait cycle), eigenvalues with a magnitude (>1) imply that the system moves away from the steady state solution after an infinitesimal perturbation, while eigenvalues (>-1 and <1) imply moving towards the steady state solution after an infinitesimal perturbation. Such a discrete system arises in gait analysis for example during a foot-strike event or when considering the Poincaré (return) map in terms of linear stability analysis (see, e.g., Wisse and Schwab, 2005). By contrast, for a continuous system ($dx/dt = A \cdot x(t)$, a differential equation describing for example the swing phase), strictly positive eigenvalues indicate that an infinitesimal perturbation leads to an exponential deviation from the current (possibly time-dependent) solution indicating the latter’s (local) instability; negative eigenvalues indicate that an infinitesimal perturbation leads to an exponential return to that solution, which in that

case is considered (locally) stable. We added a corresponding footnote when introducing the eigenvalue of discrete systems:

Page 10, footnote: “This is different from continuous systems, where the maximum eigenvalue $\lambda > 0$ indicates local instability.”

Possibly related to this, I think that the description of the “trajectory-normal divergence rate” metric would benefit from more explanation. Based on this metric, it seems that a system with two eigenvalues of +1 and -1 at some point in the gait cycle would be defined to have the same “stability” as a system with two eigenvalues of 0 and 0. This seems counterintuitive to me.

>> Thanks for posing this excellent question. The first thing to mention is that some papers (Ali and Menzinger, 1999; Norris et al., 2008) have shown that although a limit cycle may be globally stable, it is not always locally stable in the sense of maximum eigenvalue of the Jacobian matrix. In some sense, the local divergence rate (or trajectory-normal divergence rate) offers a less stringent condition than the maximum eigenvalue for defining local stability in that it only requires the sum of eigenvalues to be less than zero. Therefore, the local divergence rate and the trajectory-normal divergence rate may be used complementary to the conventional maximum eigenvalue. But consider this a conjecture in the mathematical sense. The issues of using the maximum eigenvalue are that: 1) it only applies to the response to perturbations along a single eigendirection; 2) the eigendirection corresponding to the maximum eigenvalue might be tangential to the trajectory, which will only yield a phase shift and not a deviation from the trajectory; 3) an unstable eigendirection might be stable at the subsequent instant, e.g., due to the rotation of the eigenvectors. The trajectory-normal divergence rate accounts for this: 1) it quantifies the growth of all perturbations around the unperturbed trajectory; 2) it removes dimension that corresponds to the tangential; 3) it is invariant against rotation of the normal coordinates since the growth of perturbations is evaluated only in the orthogonal space that is always normal to the time-varying tangential. To further explain this, we added the following:

Page 9, line 10 to 15: “The advantages of using the trajectory-normal divergence rate instead of the maximum eigenvalue of the Jacobian to quantify local stability are threefold: 1) it quantifies the growth of all perturbations around the unperturbed trajectory rather than in a single eigendirection; 2) it removes dimension that corresponds to the tangential; 3) it is invariant against rotation of the normal coordinates since the growth of perturbations is evaluated only in the orthogonal space that is always normal to the time-varying tangential.”

Similarly, could Figure 4 be explained in a little more detail? How many eigenvalues are plotted here? While I understand that they may be overlapping, it’s not clear from the figure that the eigenvalues sum to the trajectory-normal divergence rate.

>> We generated a new figure 5 and plotted all three individual curves of the eigenvalues in different colors and line styles to improve legibility.

Minor comments:

1) If possible given word limits, I think it would be helpful to readers to briefly define stability and robustness in the Abstract.

>> We added a definition in the abstract:

Page 1, line 17 to 23: “These measures are closely related to the often-employed maximum finite-time Lyapunov exponent and maximum Floquet multiplier that both assess a system’s response to infinitesimal perturbations... We correlated the measures with gait robustness, i.e. the largest perturbation a walker can handle.”

2) Was there a maximum of one period-one solution identified for each parameter combination? (see page 6, line 43) I seem to remember from the early McGeer simulations that some parameters produced multiple limit cycles – such as a short-period and long-period gait. Maybe these “extra” solutions were prevented by the other criteria used in this manuscript.

>> We only considered long-period gaits, therefore there is a maximum of one period-one solution identified for each parameter combination. The short-period gaits are unstable (McGeer 1990, the International Journal of Robotics Research; Garcia et al., 1998, Journal of Biomechanical Engineering), and since we ignored all the unstable solutions with a Floquet multiplier that has modulus > 1 , the short-period gaits were automatically omitted from further analysis. To clarify this, we added the following text:

Page 6, line 2 to 4: “We ignored all the unstable solutions (and thus all the short-period gaits; Mcgeer,1990) in further calculations by removing all the solutions with a maximum Floquet multiplier with a modulus >1 .”

3) In Table 1, could you explicitly state what the models are?

>> We extended the index of the three models to make the table more readable:
 $1_{\text{Euler}}, 2_{\text{Euler}}, 1_{\text{Hamiltonian}}$

4) This may just be a matter of semantics (and does not necessarily need to be addressed), but is foot-strike divergence truly a “phase-dependent stability metric”, as implied on page 11 line 49? By looking at only one point in the gait cycle, this would appear to be fairly similar to previous stability metrics that I don’t think were described as “phase-dependent”.

>> Our foot strike stability measure clearly differs from conventional Floquet multipliers. The latter defines stability of the gait cycle as a whole, whereas our foot strike measure only measures stability of the foot strike event. As the foot strike can be considered an important phase of the gait cycle, we believe it is fair to call it a phase-dependent stability measure. We have added the following sentence to make this explicit:

Page 10, line 29 to 30: “Since the foot strike event can be considered an important phase in the gait cycle, we also considered its divergence as a phase-dependent stability measure.”

Reviewer 3:

This study investigated whether different measures of phase-dependent local dynamic stability can predict gait robustness of different models of compass-walkers. The study found that for most of the measures the phase-dependent local dynamic stability was dependent on model type and on coordinate representations. The paper is well structured and well written. However, I have some major and general concerns with this study:

>> Thank you very much for the detailed criticism. Below we provide a point-by-point reply, which can hopefully clear some of your doubts.

1) The use of phase-dependent local dynamic stability for assessment of gait instabilities and prediction of falls is a quite narrow field of research with a limited number of contributions. The authors state in the abstract “Current stability measures that estimate gait stability and robustness appear limited in predicting falls in older adults” for which I completely agree. Even though there is a paper (Ihlen et al., 2015) showing increased phase-dependent instabilities for fallers compared to non-fallers, there is close to no empirical evidence for the importance of these measurements in fall prediction. Thus, I do not see the importance of testing these measures on simple models of compass-walkers without more evidence in favor of their strong predictive value.

>> We agree that there is very limited empirical evidence in favor of phase-dependent stability measures. However, we would like to draw your attention to our review paper (Bruijn et al., 2013, Royal Society Interface). There we discussed the validity of stability measures, and states that they should be: 1) theoretically valid; 2) valid in simple models, which is exactly what our manuscript aims for; 3) valid in experimental studies; 4) valid in larger cohort studies. Our manuscript’s endeavor was to test the validity of phase-dependent stability measures to predict gait robustness in simple walking models, before testing them in human walking. Although most of the results of our manuscript fail to show strong relationships between phase-dependent stability measures and robustness, it points out which measures cannot predict gait robustness well. This provides guidance for both model studies and experimental studies on humans in the future. By the same token we can identify, which measure (divergence of foot strike) predicts comparably well and direct further studies to improve it; here the keyword to mention is “check for coordinate dependency”. We are therefore convinced that our manuscript will make a valuable contribution to the literature. Upon revision we added the following text in the Introduction:

Page 3, line 12 to 14: “Still, only a limited number of studies (Fino et al., 2018; Ihlen et al., 2012, 2015, 2018; Mahmoudian et al., 2016) on human gait used such phase-dependent stability measures, and further validation is needed.”

2) It is a paradox to me, why the authors do not include the measure utilized in Ihlen et al (2015) since this is (to the reviewer’s knowledge) the only empirical evidence of phase-dependent local dynamic stability predicting falls in older persons. A major challenge of the definition used in Ihlen et al (2015) is that the phase-dependent stability refers to a percentage of the gait cycle and not to events like foot-strike and swing phase used in the authors study.

Still, the inclusion of the measure would have increased the relevance of the paper for the development of new biometrics for fall prediction.

>> The reason for not applying the method proposed by Ihlen et al. (2015, BioMed Research International) was and is that we aim for testing the merits of phase-dependent stability measures in an ‘ideal’ situation, i.e. deterministic *in silico* data, rather than using experimental data, where one certainly has to rely on advanced time series analysis methods. Ihlen et al.’s (2015) algorithm requires a noisy input, which when simulating models raises questions about the kind of noise to incorporate: should it be additive or multiplicative dynamical noise, measurements noise, should it contain (temporal) correlations, etc. Choosing one over the other will definitely influence the outcome of any subsequent analysis. While interesting, this is clearly beyond the scope of the current manuscript.

To accommodate your concerns, we added the following text in the last paragraph of Introduction:

Page 3, Line 27 to 29: “Since we aimed to test the theoretical merits of phase-dependent stability measures, we here circumvent the inclusion of time series analysis method. Hence, we did not apply the method proposed by Ihlen et al., (2012, 2015).”

3) Even though the results of present study may shed some light on why phase-dependent stability is not a sensitive predictor of falls, there is probably numerous other factors, like the problem of noise and non-stationarities affecting the numerical estimation of these measures from data, that might be more interesting for the general audience of this journal.

>> We admit that there are many interesting topics like noise and non-stationarities and their effects on stability measures. As such, our current contribution should be considered a first step in which we investigate correlations between robustness and phase-dependent stability measures in ‘ideal’ conditions and on ‘simple’ mechanical models. If correlations turn out to be low, or absent (as seems to be the case), then there will be no need to study the implementation of more advanced numerical algorithms. We clarified this as follows:

Page 3, line 19 to 21: “A positive answer will encourage application to experimental data while a negative answer should be considered a call for alternative approaches.”

In conclusion, I am struggling to see the clear scientific relevance of the present study in the important field of developing new biometrics for fall prediction. However, the study may be a more important contribution in analyzing the gait robustness of simple humanoid robots. Thus, I cannot recommend this manuscript to be published in Royal Society Open Science in its current form.

>> We regret to read this. In general, predicting falls (in particular in the elderly) is a very important issue in society. Several studies tried to find measures that predict falls experimentally in human walking without first testing them *in silico*. Frankly, we believe this is not the best approach in view of the complexity of the human locomotor system and unavoidable measurement artifacts. Many studies used unique stability measures,

suggesting a high-risk of type-1 errors (Hamacher et al., 2011, Journal of the Royal Society Interface). If feasible, one should first test potential fall predictors in tractable dynamical walking models capable of capturing key aspects of real walking. In this way, we can test and understand the (non-)validity of potential robustness predictors and use the outcome to develop time series analysis methods able to experimentally capture the same measures. The measures developed in our study have been based on theoretical arguments and recent developments in the field of dynamical system theory. Most of them turned out to have limited predictive value for gait robustness. More so, their outcome depends on the coordinate frame rendering them unreliable in more realistic/complicated settings.